# Fixing Incomplete Value Function Decomposition for Multi-Agent Reinforcement Learning

## Abstract

Value function decomposition methods for cooperative multi-agent reinforcement learning compose joint values from individual per-agent utilities, and train them using a joint objective. To ensure that the action selection process between individual utilities and joint values remains consistent, it is imperative for the composition to satisfy the *individual-global max* (IGM) property. Although satisfying IGM itself is straightforward, most existing methods (e.g., VDN, QMIX) have limited representation capabilities and are unable to represent the full class of IGM values, and the one exception that has no such limitation (QPLEX) is unnecessarily complex. In this work, we present a simple formulation of the full class of IGM values that naturally leads to the derivation of QFIX, a novel family of value function decomposition models that expand the representation capabilities of prior models via a thin "fixing" layer. We derive multiple variants of QFIX, and implement three variants in two well-known multi-agent frameworks. We perform an empirical evaluation on multiple SMACv2 and Overcooked environments, which confirms that QFIX (i) succeeds in enhancing the performance of prior methods, (ii) learns more stably and performs better than its main competitor QPLEX, and (iii) achieves this while employing the simplest and smallest mixing models.

## 1 Introduction

Centralized training for decentralized execution (CTDE) (Lowe et al., 2017; Rashid et al., 2020b; Wang et al., 2020) is a powerful framework for cooperative multi-agent reinforcement learning (MARL). CTDE is characterized by a centralized training phase where privileged information is shared freely and used holistically to train the agents, and a decentralized execution phase where agents act independently in adherence to the standard constraints of decentralized control. As a consequence of a training phase that is informed by the full team's behavior and experiences (and, when feasible, the environment state), CTDE is commonly associated with increased coordination between agents and superior performances.

Value function decomposition (Sunehag et al., 2017; Rashid et al., 2020b; Wang et al., 2020) is a class of CTDE methods that construct a joint team value from individual per-agent utilities that encode agent behaviors. By training the joint value on a joint centralized objective, the individual utilities are also indirectly trained, resulting in decentralized agent policies that can be executed independently. Since its inception, value function decomposition has become a topic of great interest in cooperative MARL, with significant research effort put in both practical algorithms (Sunehag et al., 2017; Son et al., 2019; Rashid et al., 2020a;b; Wang et al., 2020; Marchesini et al., 2024) and theoretical understanding (Wang et al., 2021; Marchesini et al., 2024). *Individual-global max* (IGM) (Son et al., 2019) has been identified as a key property that connects individual utilities and joint values, ensuring that their associated decision making processes remain consistent.

In this work, we advance both theory and practice of value function decomposition. We formulate a novel simple formulation of IGM-complete value function decomposition. Our formulation (i) correctly addresses general decentralized partially observable control (avoiding strong assumptions like full observability or centralized control), and (ii) highlights the core mechanism that characterizes the full IGM-complete function class. In contrast, prior methods fail to satisfy at least one of these criteria (usually the first, which limits the expressive capabilities and performance of models). We introduce QFIX, a novel family of value function decomposition methods inspired by our formu-

lation of IGM-complete decomposition. QFIX employs a simple "fixing" network to extend the representation capabilities of prior methods. We derive two main specializations of QFIX called QFIX-sum and QFIX-mono, respectively obtained by "fixing" VDN (Sunehag et al., 2017) and QMIX (Rashid et al., 2020b). To provide further insights into the core mechanisms that make value function decomposition so effective, we also derive QFIX-lin, a third variant that technically falls just outside of the QFIX family, but combines QFIX-sum with a core component of QPLEX. Finally, we extend prior work on state-based value function decomposition to QFIX. Empirical evaluations on the StarCraft Multi-Agent Challenge v2 (SMACv2) (Ellis et al., 2023) and Overcooked (Carroll et al., 2020) demonstrates that QFIX (i) is effective at enhancing prior non-IGM-complete methods like VDN and QMIX, (ii) is simpler to implement and understand, and require smaller models than QPLEX, a state-of-the-art method in IGM-complete value function decomposition, (iii) is competitive or outperforms QPLEX while also showing more stable convergence. An additional evaluation of model size confirms that the superior performance of QFIX is attributed to the intrinsic mixing approach rather than by augmenting baseline parameters.

## 2 RELATED WORK

Value Decomposition Networks (VDN) (Sunehag et al., 2017) are a precursor to value decomposition methods that employ a simple additive composition of individual utilities. QMIX (Rashid et al., 2020b) employs a monotonic composition that generalizes the function class of VDN resulting in significant performance improvements. Both VDN and QMIX have restricted function classes, and several methods have attempted to overcome the limits of purely additive or monotonic composition and achieve broader expressiveness. Weighted-QMIX (WQMIX) (Rashid et al., 2020a) aims to expand the function class of QMIX to non-monotonic cases so as to include optimal values $Q^*$. However, WQMIX is specifically developed for *fully observable* multi-agent environments (MMDPs), and its theory does not generalize to *partially observable* DecPOMDPs. In contrast, QFIX is fully consistent with the general case of partially observable decentralized control. Son et al. (2019) identify *individual-global max* (IGM) as a core property that corresponds to consistency between the individual and joint decision making processes. Notably, VDN and QMIX satisfy IGM, but are unable to represent the entire IGM-complete function class. QTRAN (Son et al., 2019) identifies a set of constraints that are sufficient to imply IGM, and employs auxiliary objectives that softly enforce those constraints. Son et al. (2019) argue that their constraints are also necessary for IGM under affine transformations, however they only show that one such affine transformation exists, rather than IGM being satisfied for all affine transformations. In contrast, QFIX is both sufficient and necessary to imply IGM, thus directly achieving the full IGM-complete function class. QPLEX (Wang et al., 2020) employs a dueling network decomposition and multiple layers of transformations to achieve the IGM-complete function class. However, QPLEX employs complex transformations that are superfluous in relation to its representation capabilities, and falls short of identifying the core underlying mechanism that is singularly responsible to achieve the IGM function class. In contrast, QFIX is both simpler to understand and to implement, and achieves the IGM function class with fewer smaller models. QPLEX is one instance in the space of IGM-complete models, and our work opens a path to explore other instances that can further improve performance while satisfying IGM.

## 3 BACKGROUND

### 3.1 DECENTRALIZED MULTI-AGENT CONTROL

A decentralized POMDP (Dec-POMDP) (Oliehoek & Amato, 2016) generalizes single-agent partially observable control by accounting for multiple decentralized agents acting concurrently to solve a shared cooperative task. A Dec-POMDP is defined by a tuple $\langle N, \mathcal{S}, \{\mathcal{A}_1, \ldots, \mathcal{A}_N\}, \{\mathcal{O}_1, \ldots, \mathcal{O}_N\}, p, T, R, O, \gamma \rangle$ composed of: (i) number of agents $N \geq 2$; (ii) state space $\mathcal{S}$; (iii) individual action and observation spaces $\mathcal{A}_i$ and $\mathcal{O}_i$; (iv) starting state distribution $p \in \Delta\mathcal{S}$; (v) state transition function $T \colon \mathcal{S} \times \boldsymbol{\mathcal{A}} \to \Delta\mathcal{S}$; (vi) joint observation function $O \colon \boldsymbol{\mathcal{A}} \times \mathcal{S} \to \Delta\boldsymbol{\mathcal{O}}$; (vii) joint reward function $R \colon \mathcal{S} \times \boldsymbol{\mathcal{A}} \to \mathbb{R}$; and (viii) discount factor $\gamma \in [0, 1)$. The number of agents $N$ induces a set of agent indices $\mathcal{I} \doteq [N]$. Agent behaviors are generally modeled as stochastic policies $\pi_i \colon \mathcal{H}_i \to \Delta\mathcal{A}_i$ that act based on their respective history $h_i \in \mathcal{H}_i \doteq \mathcal{O}_i \times (\mathcal{A}_i \times \mathcal{O}_i)^*$. Joint action, observation, and history spaces are defined as the respective Cartesian products $\boldsymbol{\mathcal{A}} \doteq \bigtimes_i \mathcal{A}_i$, $\boldsymbol{\mathcal{O}} \doteq \bigtimes_i \mathcal{O}_i$, and $\boldsymbol{\mathcal{H}} \doteq \bigtimes_i \mathcal{H}_i$. Therefore, joint actions

$\boldsymbol{a} = (a_1, \ldots, a_N)$, observations $\boldsymbol{o} = (o_1, \ldots, o_N)$, and histories $\boldsymbol{h} = (h_1, \ldots, h_N)$ are tuples of the respective individual actions, observations, and histories. The combined behavior of all policies is represented as a joint (but still decentralized) policy $\boldsymbol{\pi}(\boldsymbol{h}, \boldsymbol{a}) \doteq \prod_i \pi_i(h_i, a_i)$ that factorizes accordingly. Decentralized multi-agent control aims to find independent policies that jointly maximize the expected sum of discounted rewards $J^{\boldsymbol{\pi}} \doteq \mathbb{E}\left[\sum_t \gamma^t R(s_t, \boldsymbol{a}_t)\right]$.

We focus on approaches that model policies implicitly via parametric utilities $\hat{Q}_i \colon \mathcal{H}_i \times \mathcal{A}_i \to \mathbb{R}$, typically via ($\epsilon$-)greedy action selection. Individual utilities can be decomposed into corresponding values $\hat{V}_i(h_i) \doteq \max_{a_i} \hat{Q}_i(h_i, a_i)$ and advantages $\hat{A}_i(h_i, a_i) \doteq \hat{Q}_i(h_i, a_i) - \hat{V}_i(h_i)$. When convenient, we employ shorthand notation for individual values $q_i \doteq \hat{Q}_i(h_i, a_i)$, $v_i \doteq \hat{V}_i(h_i)$, and $u_i \doteq \hat{A}_i(h_i, a_i)$, and their joint tuples $\boldsymbol{q} \doteq (q_1, \ldots, q_N)$, $\boldsymbol{v} \doteq (v_1, \ldots, v_N)$, and $\boldsymbol{u} \doteq (u_1, \ldots, u_N)$.

## 3.2 VALUE FUNCTION DECOMPOSITION

Value function decomposition methods (Sunehag et al., 2017; Rashid et al., 2020b; Wang et al., 2020) construct joint values $\hat{Q}(\boldsymbol{h}, \boldsymbol{a})$ from individual per-agent *utilities* $\hat{Q}_i(h_i, a_i)$. We specifically use the term *utility* to underscore the fact that $\hat{Q}_i(h_i) \in \mathbb{R}^{\mathcal{A}_i}$ merely represents an ordering over actions, rather than any notion of expected performance. Notably, $\hat{Q}_i$ is *not* directly trained for policy evaluation or optimization, and neither $\hat{Q}_i(h_i, a_i) \approx Q_i^{\boldsymbol{\pi}}(h_i, a_i)$ nor $\hat{Q}_i(h_i, a_i) \approx Q_i^*(h_i, a_i)$ are expected interpretations of well-trained utilities.

Value function decomposition methods employ joint models $\hat{Q}(\boldsymbol{h}, \boldsymbol{a})$ that are a function of the individual utilities $\hat{Q}_i(h_i, a_i)$, and mainly differ in terms of the relationship that is enforced and the corresponding emergent properties. The joint model $\hat{Q}(\boldsymbol{h}, \boldsymbol{a})$ is trained on a *joint* objective that optimizes the joint values and behavior, and indirectly trains the individual utilities and behaviors,

$$\mathcal{L}_{\hat{Q}}(\boldsymbol{h}, \boldsymbol{a}, r, \boldsymbol{o}) \doteq \frac{1}{2}\left(r + \gamma \max_{\boldsymbol{a}'} \hat{Q}^-(\boldsymbol{h}\boldsymbol{a}\boldsymbol{o}, \boldsymbol{a}') - \hat{Q}(\boldsymbol{h}, \boldsymbol{a})\right)^2. \tag{1}$$

**Individual-global max**  Son et al. (2019) identify individual-global max (IGM) as a useful property of decomposition models to achieve decentralized action selection and address scaling concerns.

**Definition 1** (Individual-Global Max). *Individual utilities* $\{Q_i(h_i, a_i)\}_{i \in \mathcal{I}}$ *and joint values* $Q(\boldsymbol{h}, \boldsymbol{a})$ *satisfy* individual-global max *(IGM) iff* $\bigtimes_i \operatorname{argmax}_{a_i} Q_i(h_i, a_i) = \operatorname{argmax}_{\boldsymbol{a}} Q(\boldsymbol{h}, \boldsymbol{a})$.[1]

IGM denotes whether the individual and global decision making processes are equivalent, and reduces the complexity of finding the maximal joint action from exponential to linear in the number of agents: For a given joint history $\boldsymbol{h}$, the full search over the joint action space $\mathcal{A}$ can be replaced with $N$ independent searches over the individual action spaces $\mathcal{A}_i$. VDN and QMIX are well-known models that satisfy IGM; however, their function classes do not span the full class of IGM values.

**Definition 2** (IGM Function Class). *We say a function class of individual utilities* $\{Q_i(h_i, a_i)\}_{i \in \mathcal{I}}$ *and joint values* $Q(\boldsymbol{h}, \boldsymbol{a})$ *is IGM-complete if it contains all and only functions that satisfy IGM.*

**VDN: additive decomposition**  Value Decomposition Network (VDN) (Sunehag et al., 2017) is a precursor to value function decomposition that uses a simple sum $\hat{Q}_{\mathrm{VDN}}(\boldsymbol{h}, \boldsymbol{a}) \doteq \sum_i \hat{Q}_i(h_i, a_i)$.

**QMIX: monotonic decomposition**  QMIX (Rashid et al., 2020b) constructs joint values as a *monotonic* function of individual utilities, $\hat{Q}_{\mathrm{MIX}}(\boldsymbol{h}, \boldsymbol{a}) \doteq f_{\mathrm{mono}}(q_1, \ldots, q_N)$, with $f_{\mathrm{mono}} \colon \mathbb{R}^N \to \mathbb{R}$ a parametric mixing network that satisfies monotonicity, $\partial_{q_i} f_{\mathrm{mono}} \geq 0$. Although the monotonic composition of QMIX generalizes VDN, it still falls short of the full IGM function class.

**QPLEX: IGM-complete decomposition**  QPLEX (Wang et al., 2020) reframes IGM in terms of advantages, and employs dueling network decomposition to the IGM function class. Given utilities $Q_i(h_i, a_i)$ and joint action-values $Q(\boldsymbol{h}, \boldsymbol{a})$, corresponding values and advantages are defined,

$$V_i(h_i) \doteq \max_{a_i} Q_i(h_i, a_i), \qquad\qquad A_i(h_i, a_i) \doteq Q_i(h_i, a_i) - V_i(h_i), \tag{2}$$

$$V(\boldsymbol{h}) \doteq \max_{\boldsymbol{a}} Q(\boldsymbol{h}, \boldsymbol{a}), \qquad\qquad A(\boldsymbol{h}, \boldsymbol{a}) \doteq Q(\boldsymbol{h}, \boldsymbol{a}) - V(\boldsymbol{h}). \tag{3}$$

---

[1] We employ set notation and Cartesian products to highlight that maximal actions may not be unique.

Wang et al. (2020) reformulate IGM as a set of constraints between individual and joint advantages.

**Proposition 1** (Advantage Constraints). *Individual utilities* $\{Q_i(h_i, a_i)\}_{i \in \mathcal{I}}$ *and joint values* $Q(\boldsymbol{h}, \boldsymbol{a})$ *satisfy IGM iff,* $\forall \boldsymbol{h} \in \mathcal{H}$, $\forall \boldsymbol{a}^* \in \mathcal{A}^*(\boldsymbol{h})$, *and* $\forall \boldsymbol{a} \in \mathcal{A} \setminus \mathcal{A}^*(\boldsymbol{h})$,

$$A(\boldsymbol{h}, \boldsymbol{a}^*) = 0\,, \qquad\qquad A_i(h_i, a_i^*) = 0\,, \qquad\qquad (4)$$

$$A(\boldsymbol{h}, \boldsymbol{a}) < 0\,, \qquad\qquad A_i(h_i, a_i) \le 0\,, \qquad\qquad (5)$$

*where* $\mathcal{A}^*(\boldsymbol{h}) \doteq \operatorname{argmax}_{\boldsymbol{a}} Q(\boldsymbol{h}, \boldsymbol{a})$ *is the set of maximal joint actions according to the joint values.*

QPLEX employs a mixing structure that enforces Proposition 1. Individual utilities $\hat{Q}_i(h_i, a_i)$ are decomposed into $\hat{V}_i(h_i)$ and $\hat{A}_i(h_i, a_i)$ and transformed using centralized information,

$$\hat{V}_i(\boldsymbol{h}) \doteq w_i(\boldsymbol{h})\hat{V}_i(h_i) + b_i(\boldsymbol{h})\,, \qquad\qquad \hat{A}_i(\boldsymbol{h}, a_i) \doteq w_i(\boldsymbol{h})\hat{A}_i(h_i, a_i)\,, \qquad (6)$$

where $w_i \colon \mathcal{H} \to \mathbb{R}_{>0}$ are parametric positive weights and $b_i \colon \mathcal{H} \to \mathbb{R}$ are parametric biases. These transformed values are aggregated as weighted sums,

$$\hat{V}_{\mathrm{PLEX}}(\boldsymbol{h}) \doteq \sum_i \hat{V}_i(\boldsymbol{h})\,, \qquad\qquad \hat{A}_{\mathrm{PLEX}}(\boldsymbol{h}, \boldsymbol{a}) \doteq \sum_i \lambda_i(\boldsymbol{h}, \boldsymbol{a})\hat{A}_i(\boldsymbol{h}, a_i)\,, \qquad (7)$$

where $\lambda_i \colon \mathcal{H} \times \mathcal{A} \to \mathbb{R}_{>0}$ are parametric positive weights. The QPLEX joint values are obtained by recombining aggregate values and advantages, $\hat{Q}_{\mathrm{PLEX}}(\boldsymbol{h}, \boldsymbol{a}) \doteq \hat{V}_{\mathrm{PLEX}}(\boldsymbol{h}) + \hat{A}_{\mathrm{PLEX}}(\boldsymbol{h}, \boldsymbol{a})$.

This sequence of decomposition, transformations, and recomposition, combined with positive weights $w_i$ and $\lambda_i$, results in the constraint from Proposition 1 being satisfied. Consequently, Wang et al. (2020) appeal to the universal approximation theorem (UAT) to argue that the function class of QPLEX is IGM-complete. In Appendix A, we address technical concerns and conclude that, based on a *weaker* form of UAT, the function class realizable by QPLEX is that of *measurable* IGM values.

**State-based value function decomposition** Practical implementations of value function decomposition methods often employ state-based joint values $Q(\boldsymbol{h}, s, \boldsymbol{a})$ and diverge from the stateless theoretical derivations in ways that may undermine core IGM properties, e.g., as seen for QMIX in `Pymarl` Rashid et al. (2020b), QMIX in `Pymarl2` Ellis et al. (2023), and both QMIX and QPLEX in `JaxMARL` Rutherford et al. (2024)) To address the effects of state in value function decomposition, Marchesini et al. (2024) formulate a state-compliant version of IGM.

**Definition 3** (State-based IGM). *Individual utilities* $\{Q_i(h_i, a_i)\}_{i \in \mathcal{I}}$ *and state-based joint values* $Q(\boldsymbol{h}, s, \boldsymbol{a})$ *satisfy state-based IGM iff* $\bigtimes_i \operatorname{argmax}_{a_i} Q_i(h_i, a_i) = \operatorname{argmax}_{\boldsymbol{a}} \mathbb{E}_{s|\boldsymbol{h}}[Q(\boldsymbol{h}, s, \boldsymbol{a})]$.

Marchesini et al. (2024) show that the state-based implementations of QMIX and QPLEX continue to satisfy IGM, while the state-based implementation of QPLEX (which employs historyless state-based weights $w_i(s), \lambda_i(s, \boldsymbol{a})$) fails to achieve the full IGM function class. Nonetheless, state-based implementations often perform well in practice, and remain a common occurrence.

## 4 FIXING INCOMPLETE VALUE FUNCTION DECOMPOSITION

Although QPLEX is IGM-complete, it is expressed as a convoluted sequence of transformations that are never fully motivated or justified. Fully unrolling the QPLEX values in terms of the individual utilities, we get $\hat{Q}_{\mathrm{PLEX}}(\boldsymbol{h}, \boldsymbol{a}) = \sum_i w_i(\boldsymbol{h})\hat{V}_i(h_i) + b_i(\boldsymbol{h}) + w_i(\boldsymbol{h})\lambda_i(\boldsymbol{h}, \boldsymbol{a})\hat{A}_i(h_i, a_i)$, a complex expression that raises questions about which components are truly important or necessary, e.g., the product of individual advantages with two types of positive weights $w_i(\boldsymbol{h})$ and $\lambda_i(\boldsymbol{h}, \boldsymbol{a})$ appears to be redundant. Ultimately, QPLEX only represents one instance in the space of all IGM-complete models, and whether simpler or better-performing models exist remains an open question.

The convoluted nature of the QPLEX transformations motivate us to find a simpler and more general formulation of IGM-complete decomposition. In this section, we first present a simple formulation of the IGM-complete function class. Then, we use this formulation to derive QFIX, a novel family of value function decomposition models that operate by "fixing" (read: expanding) the representation capabilities of prior non-IGM-complete models. We derive two primary instances of QFIX based on "fixing" VDN and QMIX respectively, and a third instance designed to resemble QPLEX. Then, we derive *additive* QFIX (Q+FIX), a simple variant of QFIX that achieves significant practical performance gains, and derive Q+FIX counterparts of the QFIX instances. Finally, we discuss state-based variants of QFIX and how the use of centralized state information affects its theoretical properties.

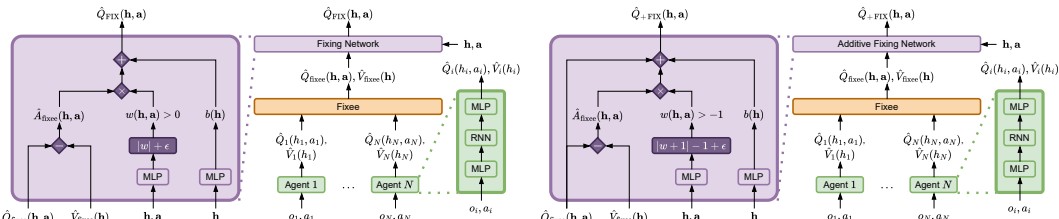

Figure 1: Diagrams for QFIX (left) and Q+FIX (right).

## 4.1 A SIMPLE PARAMETERIZATION OF THE IGM FUNCTION CLASS

We aim to formalize IGM-complete value function decomposition in its simplest and most essential form. We begin by simplifying Proposition 1, noting that three of its four constraints are satisfied by definition; the only one that requires active enforcement is $A_i(h_i, a_i^*) = 0$, or equivalently $A(\boldsymbol{h}, \boldsymbol{a}) = 0 \implies \forall i \, (A_i(h_i, a_i) = 0)$. However, we also note that Proposition 1 is actually underspecified, and misidentifies the case where $\forall i (A_i(h_i, a_i) = 0)$ and $A(\boldsymbol{h}, \boldsymbol{a}) < 0$ as compliant with IGM when it is not.[2] To address this case, we need $\forall i \, (A_i(h_i, a_i) = 0) \implies A(\boldsymbol{h}, \boldsymbol{a}) = 0$.

**Proposition 2** (Simplified Advantage Constraints). *Individual utilities* $\{Q_i(h_i, a_i)\}_{i \in \mathcal{I}}$ *and joint values* $Q(\boldsymbol{h}, \boldsymbol{a})$ *satisfy IGM iff* $\forall i \, (A_i(h_i, a_i) = 0) \iff A(\boldsymbol{h}, \boldsymbol{a}) = 0$, *or equivalently, via contraposition,* $\exists i \, (A_i(h_i, a_i) < 0) \iff A(\boldsymbol{h}, \boldsymbol{a}) < 0$.

In essence, constructing joint advantages that are negative iff any of the individual advantages are negative is both sufficient and necessary to satisfy IGM. Consider the purposefully named function

$$Q_{\text{IGM}}(\boldsymbol{h}, \boldsymbol{a}) \doteq w(\boldsymbol{h}, \boldsymbol{a}) f(u_1, \dots, u_N) + b(\boldsymbol{h}) \,, \tag{8}$$

where $w \colon \mathcal{H} \times \mathcal{A} \to \mathbb{R}_{>0}$ is an arbitrary positive function of joint history and joint action, $b \colon \mathcal{H} \to \mathbb{R}$ is an arbitrary function of joint history, and $f \colon \mathbb{R}_{\leq 0}^N \to \mathbb{R}_{\leq 0}$ is a non-positive function that is zero iff all inputs are zero, e.g., $f(u_1, \dots, u_N) = \sum_i u_i$ is a simple instance of $f$. Then,

$$V_{\text{IGM}}(\boldsymbol{h}) \doteq \max_{\boldsymbol{a}} Q_{\text{IGM}}(\boldsymbol{h}, \boldsymbol{a}) = b(\boldsymbol{h}) \,, \tag{9}$$

$$A_{\text{IGM}}(\boldsymbol{h}, \boldsymbol{a}) \doteq Q_{\text{IGM}}(\boldsymbol{h}, \boldsymbol{a}) - V_{\text{IGM}}(\boldsymbol{h}) = w(\boldsymbol{h}, \boldsymbol{a}) f(u_1, \dots, u_N) \,. \tag{10}$$

Essentially, $Q_{\text{IGM}}$ denotes a relationship where any deviation from individual maximality (characterized by at least one negative utility $u_i < 0$, and corresponding to a negative $f(u_1, \dots, u_N) < 0$) is transformed into an arbitrary deviation $w(\boldsymbol{h}, \boldsymbol{a}) f(u_1, \dots, u_N) < 0$ from joint maximality (and vice versa). Per Proposition 2, $Q_{\text{IGM}}$ represents the full IGM function class.

**Proposition 3.** *For any* $f$, $w$, *and* $b$, *values* $\{Q_i\}_{i \in \mathcal{I}}$ *and* $Q_{\text{IGM}}$ *satisfy IGM. For any* $f$, *and given free choice of* $w$ *and* $b$, *the function class of* $\{Q_i\}_{i \in \mathcal{I}}$ *and* $Q_{\text{IGM}}$ *is IGM-complete. (Proof in Appendix B.1.)*

$Q_{\text{IGM}}$ is a simple formulation of the IGM function class based on a single weighted (via $w$) transformation (via $f$) of individual advantages. Next, we explore how this formulation directly inspires the derivation of QFIX, a closely related novel family of value function decomposition models.

## 4.2 QFIX

Let $\hat{Q}_{\text{fixee}}(\boldsymbol{h}, \boldsymbol{a})$ denote a "fixee" value function decomposition model that satisfies IGM but is not IGM-complete, e.g., VDN or QMIX. Equation (8) suggests a method to "fix" $\hat{Q}_{\text{fixee}}$ and expand its function class to match the full class of IGM functions. We can extend the expressiveness of $\hat{Q}_{\text{fixee}}$ by processing it through a "fixing" network that resembles Eq. (8),

$$\hat{Q}_{\text{FIX}}(\boldsymbol{h}, \boldsymbol{a}) \doteq w(\boldsymbol{h}, \boldsymbol{a}) \hat{A}_{\text{fixee}}(\boldsymbol{h}, \boldsymbol{a}) + b(\boldsymbol{h}) \,, \tag{11}$$

---

[2]Luckily, this issue is exclusive to Proposition 1, and QPLEX itself does not suffer from the same issue.

where $w\colon \mathcal{H} \times \mathcal{A} \to \mathbb{R}_{>0}$ is a parametric positive model, $b\colon \mathcal{H} \to \mathbb{R}$ is a parametric model, and $\hat{A}_{\text{fixee}}\colon \mathcal{H} \times \mathcal{A} \to \mathbb{R}_{\leq 0}$ is the non-positive joint advantage of the fixee as defined by

$$\hat{V}_{\text{fixee}}(\boldsymbol{h}) \doteq \max_{\boldsymbol{a}} \hat{Q}_{\text{fixee}}(\boldsymbol{h}, \boldsymbol{a}), \qquad \hat{A}_{\text{fixee}}(\boldsymbol{h}, \boldsymbol{a}) \doteq \hat{Q}_{\text{fixee}}(\boldsymbol{h}, \boldsymbol{a}) - \hat{V}_{\text{fixee}}(\boldsymbol{h}). \qquad (12)$$

See Fig. 1 for a diagram of QFIX. We note that $\hat{A}_{\text{fixee}}(\boldsymbol{h}, \boldsymbol{a}) = 0$ iff the joint action $\boldsymbol{a}$ is maximal according to $\hat{Q}_{\text{fixee}}$, and negative otherwise. Given that $\hat{Q}_{\text{fixee}}$ satisfies IGM by assumption, $\boldsymbol{a}$ is maximal iff the individual actions $a_i$ are maximal according to $\hat{Q}_i(h_i, a_i)$, or, equivalently, iff $\hat{A}_i(h_i, a_i) = 0$. In short, $\hat{A}_{\text{fixee}}(\boldsymbol{h}, \boldsymbol{a})$ satisfies the requirements of $f$ under Eq. (8).

**Proposition 4.** *QFIX satisfies IGM. The function class of QFIX is that of (measurable) IGM values. (Proof in Appendix B.2.)*

Given the free choice of fixee model $\hat{Q}_{\text{fixee}}$, QFIX really represents a wide family of value function decomposition models. This allows us to consider more or less complex fixees (e.g., VDN, QMIX) and explore various possible tradeoffs between minimizing the complexity of the fixee model and minimizing the "fixing" burden on the fixing models $w, b$. In our empirical evaluation, we will generally find that the fixing burden on $w, b$ is not significant, and that it is perfectly reasonable to combine QFIX with simpler fixees like VDN or *tiny* versions of parametric fixees like QMIX.

**Relationship to QPLEX**  The advantage component of QFIX, $w(\boldsymbol{h}, \boldsymbol{a})\hat{A}_{\text{fixee}}(\boldsymbol{h}, \boldsymbol{a})$, is similar to one of the transformations of QPLEX, $\sum_i \lambda_i(\boldsymbol{h}, \boldsymbol{a})\hat{A}_i(\boldsymbol{h}, a_i)$, which comparably applies positive weights to transformed aggregates of the individual advantages. This similarity is no coincidence, as it is specifically that component of QPLEX that is responsible for achieving IGM-completeness; it is a more convoluted form of our proposed fixing structure. However, QPLEX also employs various other transformations that do not contribute to the IGM-complete function class, and their necessity remains questionable (beyond general considerations of modeling structure and size).

The weights $\lambda_i(\boldsymbol{h}, \boldsymbol{a})$ employed by QPLEX are also more complex in that there is one such model per agent, and each is implemented via self-importance. In contrast, we employ a simpler structure based on a single model implemented as a simple feed-forward network, and still manage to achieve performance improvements. Our formulation is simpler in that it focuses entirely on this single transformation, which is minimally sufficient to guarantee IGM-completeness.

**Fixing VDN**  We define QFIX-sum as an instance of QFIX based on "fixing" VDN, i.e., with $\hat{Q}_{\text{fixee}}(\boldsymbol{h}, \boldsymbol{a}) = \hat{Q}_{\text{VDN}}(\boldsymbol{h}, \boldsymbol{a})$, which results in (see Appendix C.3 for an explicit derivation)

$$\hat{Q}_{\text{FIX-sum}}(\boldsymbol{h}, \boldsymbol{a}) = w(\boldsymbol{h}, \boldsymbol{a})\sum_i \hat{A}_i(h_i, a_i) + b(\boldsymbol{h}). \qquad (13)$$

**Fixing QMIX**  We define QFIX-mono as an instance of QFIX based on "fixing" QMIX, i.e., with $\hat{Q}_{\text{fixee}}(\boldsymbol{h}, \boldsymbol{a}) = \hat{Q}_{\text{MIX}}(\boldsymbol{h}, \boldsymbol{a})$, which results in (see Appendix C.4 for an explicit derivation)

$$\hat{Q}_{\text{FIX-mono}}(\boldsymbol{h}, \boldsymbol{a}) = w(\boldsymbol{h}, \boldsymbol{a})\left(f_{\text{mono}}(q_1, \ldots, q_N) - f_{\text{mono}}(v_1, \ldots, v_N)\right) + b(\boldsymbol{h}). \qquad (14)$$

**Simplifying QPLEX**  Given the discussed similarity between QFIX and QPLEX, we may consider another variant of QFIX that also applies per-agent positive weights $w_i(\boldsymbol{h}, \boldsymbol{a}) > 0$. Due to the linear structure that generalizes the additive structure of QFIX-sum, we call this variant QFIX-lin.

$$\hat{Q}_{\text{FIX-lin}}(\boldsymbol{h}, \boldsymbol{a}) \doteq \sum_i w_i(\boldsymbol{h}, \boldsymbol{a})\hat{A}_i(h_i, a_i) + b(\boldsymbol{h}). \qquad (15)$$

QFIX-lin does not strictly satisfy the form of Eq. (11), however, it represents a close enough variant of QFIX-sum that we consider it QFIX-adjacent and name it accordingly. QFIX-lin is a strict generalization of QFIX-sum, which can be recovered as a special case where all the weights $w_i(\boldsymbol{h}, \boldsymbol{a})$ are equal. Formally, we must prove the IGM properties of QFIX-lin separately.

**Proposition 5.** *QFIX-lin satisfies IGM. The function class of QFIX-lin is that of (measurable) IGM values. (Proof in Appendix B.3.)*

**Recovering the fixee model**  We note that QFIX is able recover the fixee model via $w(\boldsymbol{h}, \boldsymbol{a}) = 1$ and $b(\boldsymbol{h}) = \hat{V}_{\text{fixee}}(\boldsymbol{h})$, for which $\hat{Q}_{\text{FIX}}(\boldsymbol{h}, \boldsymbol{a}) = \hat{A}_{\text{fixee}}(\boldsymbol{h}, \boldsymbol{a}) + \hat{V}_{\text{fixee}}(\boldsymbol{h}) = \hat{Q}_{\text{fixee}}(\boldsymbol{h}, \boldsymbol{a})$. Such values of $w(\boldsymbol{h}, \boldsymbol{a})$ and $b(\boldsymbol{h})$ establish a direct relationship between the fixee and fixed models, which is relevant as we next use this relationship to derive a better-performing *additive* variant of QFIX.

## 4.3 ADDITIVE QFIX (Q+FIX)

In this section, we further derive a simple reparameterization of QFIX which, albeit having the same theoretical properties, achieves significant practical performance improvements. This variant takes on an additive form when compared to the fixee model, hence its name *additive QFIX* (Q+FIX).

As previously noted, the values of $w(\boldsymbol{h}, \boldsymbol{a}) = 1$ and $b(\boldsymbol{h}) = \hat{V}_{\text{fixee}}(\boldsymbol{h})$ hold a special significance for QFIX. Q+FIX is obtained by reparameterizing $w$ and $b$ to incorporate such values additively,

$$\hat{Q}_{\text{+FIX}}(\boldsymbol{h}, \boldsymbol{a}) \doteq (w(\boldsymbol{h}, \boldsymbol{a}) + 1)\hat{A}_{\text{fixee}}(\boldsymbol{h}, \boldsymbol{a}) + (b(\boldsymbol{h}) + \hat{V}_{\text{fixee}}(\boldsymbol{h}))$$
$$= \hat{Q}_{\text{fixee}}(\boldsymbol{h}, \boldsymbol{a}) + w(\boldsymbol{h}, \boldsymbol{a})\hat{A}_{\text{fixee}}(\boldsymbol{h}, \boldsymbol{a}) + b(\boldsymbol{h}) \,, \tag{16}$$

where $w \colon \boldsymbol{\mathcal{H}} \times \boldsymbol{\mathcal{A}} \to \mathbb{R}_{>-1}$ is a parametric model constrained by $w(\boldsymbol{h}, \boldsymbol{a}) > -1$, $b \colon \boldsymbol{\mathcal{H}} \to \mathbb{R}$ is a parametric model, and $\hat{Q}_{\text{fixee}}$ and $\hat{A}_{\text{fixee}}$ are the fixee action-values and advantages. Note that, with the reparameterization of $w$, its constraint has changed; Since $w(\boldsymbol{h}, \boldsymbol{a}) + 1 > 0$ must satisfy the positivity constraint from QFIX, the corresponding constraint for Q+FIX is therefore $w(\boldsymbol{h}, \boldsymbol{a}) > -1$.

See Fig. 1 for a diagram. This reparameterization allows Q+FIX to more directly exploit the original form of the fixee model, extending its representation via a separate additive component $\Delta(\boldsymbol{h}, \boldsymbol{a}) \doteq w(\boldsymbol{h}, \boldsymbol{a})\hat{A}_{\text{fixee}}(\boldsymbol{h}, \boldsymbol{a}) + b(\boldsymbol{h})$ we call the *fixing intervention*. Because Q+FIX is a simple reparameterization of QFIX, the results from Propositions 4 and 5 apply trivially to their Q+FIX counterparts. Next, we look at specific instances and other relevant implementation details.

**Q+FIX-{sum,mono,lin}**  The Q+FIX counterparts to QFIX-{sum,mono,lin} are as follows. See Appendices C.5 to C.7 for their corresponding derivations and specialized diagrams.

$$\hat{Q}_{\text{+FIX-sum}}(\boldsymbol{h}, \boldsymbol{a}) = \sum_i \hat{Q}_i(h_i, a_i) + w(\boldsymbol{h}, \boldsymbol{a})\sum_i \hat{A}_i(h_i, a_i) + b(\boldsymbol{h}) \,, \tag{17}$$

$$\hat{Q}_{\text{+FIX-mono}}(\boldsymbol{h}, \boldsymbol{a}) = f_{\text{mono}}(\boldsymbol{q}) + w(\boldsymbol{h}, \boldsymbol{a})\left(f_{\text{mono}}(\boldsymbol{q}) - f_{\text{mono}}(\boldsymbol{v})\right) + b(\boldsymbol{h}) \,, \tag{18}$$

$$\hat{Q}_{\text{+FIX-lin}}(\boldsymbol{h}, \boldsymbol{a}) = \sum_i \hat{Q}_i(h_i, a_i) + \sum_i w_i(\boldsymbol{h}, \boldsymbol{a})\hat{A}_i(h_i, a_i) + b(\boldsymbol{h}) \,. \tag{19}$$

**Detaching the advantages**  The additive form of Q+FIX enables the use of an implementation detail already employed by QPLEX that significantly improves performance: the detachment of the advantages when computing gradients. This can be expressed using the stop-gradient operator,[3]

$$\hat{Q}_{\text{+FIX}}(\boldsymbol{h}, \boldsymbol{a}) = \hat{Q}_{\text{fixee}}(\boldsymbol{h}, \boldsymbol{a}) + w(\boldsymbol{h}, \boldsymbol{a})\,\text{stop}[\hat{A}_{\text{fixee}}(\boldsymbol{h}, \boldsymbol{a})] + b(\boldsymbol{h}) \,. \tag{20}$$

The reason why detaching the advantages improves performance is not fully understood. Wang et al. (2020, Appendix B.2) argue that it (cit.) "*increases the optimization stability of the max operator of the dueling structure*", in reference to dueling networks (Wang et al., 2016). However, the connection between the detach and dueling networks remains unclear. Instead, we hypothesize that detaching the advantage may mitigate adverse effects that the fixing structure may have on the gradients $\nabla_{\theta_i}\hat{Q}_{\text{+FIX}}(\boldsymbol{h}, \boldsymbol{a})$ of the joint values w.r.t. the agent parameters $\theta_i$ (see Appendix D).

**Annealing the intervention**  In an effort to stabilize the learning, we have found it occasionally useful to introduce the fixing intervention smoothly during the early stages of training ($\approx 5\%$ of total timesteps) by employing an auxiliary loss $\lambda_\Delta \cdot \Delta^2(\boldsymbol{h}, \boldsymbol{a})$ that minimizes the squared intervention, with a weight $\lambda_\Delta$ that is annealed from a starting value down to 0, ultimately disabling this intervention loss. This likely ensures that the early stages of training are focused more on training the fixee values, so that the fixing intervention can later on focus on making smaller detailed adjustments.

---

[3]The stop-gradient function is a mathematical anomaly whose value behaves like the identity function, $\text{stop}[x] = x$, while its gradient behaves like the zero function, $\nabla_x \text{stop}[x] = 0$. It is a functionality commonly provided by deep learning frameworks, e.g., `pytorch` provides this via the `Tensor.detach()` method.

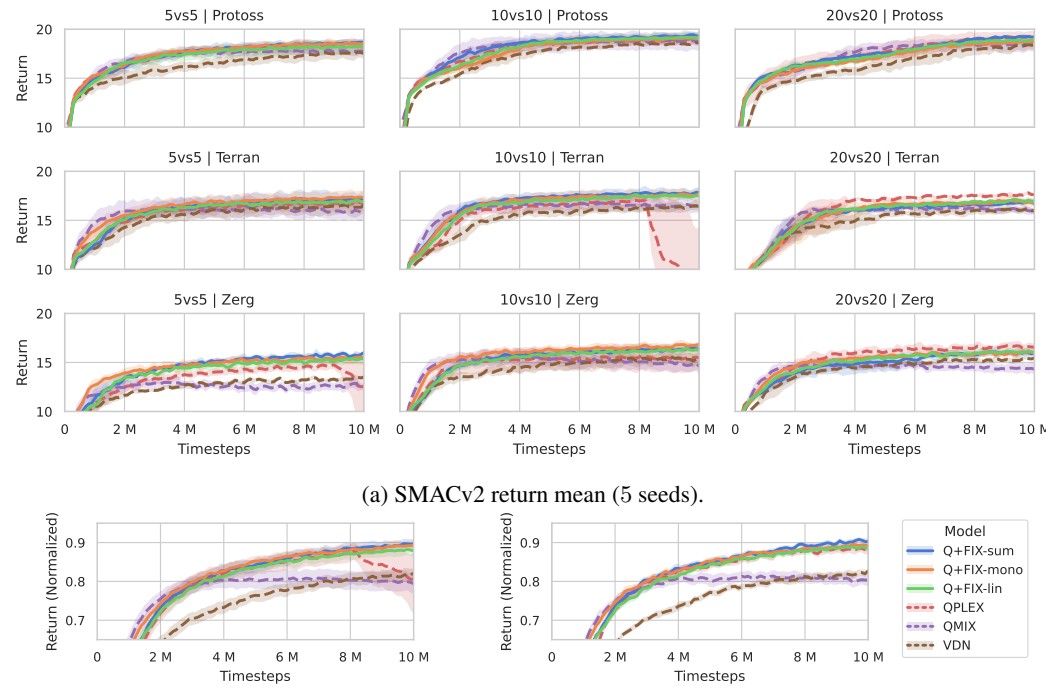

(a) SMACv2 return mean (5 seeds).

(b) SMACv2 return mean aggregate (45 seeds).    (c) SMACv2 return IQM aggregate (45 seeds).

Figure 2: SMACv2 results, bootstrapped $95\%$ CI. Aggregate returns are normalized per-task via $\tilde{G}_i \doteq \frac{G_i - \min_k G_k}{\max_k G_k - \min_k G_k}$, where $\{G_i\}_i$ is the total set of returns logged by all models in a given task.

### 4.4 STATE-BASED VARIANTS

As with QMIX and QPLEX, we may consider state-based variants of QFIX that partially deviate from the stateless theory developed so far. Such variants warrant an explicit discussion on the implications of employing centralized state information (Marchesini et al., 2024). Different versions of state-based QFIX are possible by combining stateless/state-based fixees with stateless/state-based fixing networks. As Q+FIX is a simple reparameterization of QFIX, its properties w.r.t the use of state are the same. We briefly summarize the conclusions for two main state-based variants of QFIX, which are comparable to those for state-based QPLEX (Marchesini et al., 2024): (History-State QFIX) When employing history-state fixing models $w(\boldsymbol{h}, s, \boldsymbol{a})$ and $b(\boldsymbol{h}, s)$, QFIX both satisfies IGM and achieves a form of IGM-complete function class. (State-Only QFIX) When employing state-only fixing models $w(s, \boldsymbol{a})$ and $b(s)$, QFIX continues to satisfy IGM, but fails to achieve the IGM-complete function class. See additional discussion in Appendix E.

## 5 EVALUATION

We perform an empirical evaluation of Q+FIX in two popular multi-agent frameworks, `Pymarl2` and `JaxMARL`. Appendices G and H contains practical details on architectures and used resources.

**SMACv2** `Pymarl2` provides baseline implementations for SMACv2 Ellis et al. (2023), a popular benchmark for cooperative multi-agent control based on the real-time strategy game StarCraft II. SMACv2 features two battling teams composed by configurable races, race-dependent and stochastically determined unit types, and team sizes. Our empirical evaluation is based on 9 scenarios obtained by combining the 3 races (`Protoss`, `Terran`, and `Zerg`) with 3 team sizes (`5vs5`, `10vs10`, and `20vs20`). We use shorthand labels, e.g., `P5`, `T10`, `Z20`. `Pymarl2` provides base implementations for VDN, QMIX, and QPLEX, and we implemented Q+FIX-{sum,mono,lin}.

Fig. 2a contains the evaluation results based on mean performance, with 5 independent runs per model per scenario. As expected, VDN fails to be a competitive baseline on its own accord. Fixing VDN via Q+FIX-sum, we are able to overcome this limitation, as noted by the corresponding performance gap. QMIX sometimes exhibits fast initial learning speeds, albeit often to a sub-competitive

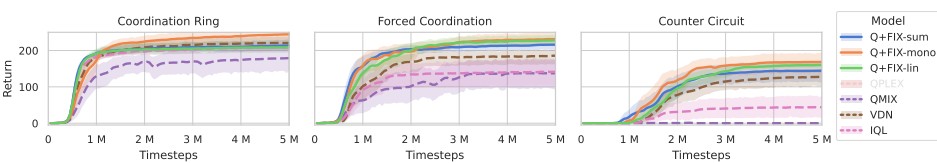

Figure 3: Overcooked return mean, bootstrapped $95\%$ CI (20 seeds).

final performance (`P5`, `T5`, `T10`, `Z10`, `T20`, `Z20`). Fixing QMIX via Q+FIX-mono, we are often able to exploit the initial learning speed and complement it with improved convergence performance. QPLEX is highly competitive and performs very well in some scenarios (`P5`, `P20`, `T20`, `Z20`), but underperforms in others (`T5`, `P10`, `Z10`), and exhibits troubling instabilities (`Z5`, `T10`). Q+FIX-lin avoids such convergence instabilities, likely as a consequence of the simpler structure. Although Q+FIX-{sum,mono,lin} generally achieve similar performances, Q+FIX-sum may be slightly out-performing other variants in some scenarios (`T5`, `Z5`), possibly an indication that a simpler compositions are not just sufficient but possibly preferable.

In accordance to the methodology suggested by Agarwal et al. (2021) to improve statistical signif-icance and alleviate the impact of outliers, Figs. 2b and 2c contain (normalized) aggregate results based on mean and interquantile mean (IQM). Even ignoring the unstable convergence of QPLEX via the aggregate IQM results, it is clear that the Q+FIX variants continue to outperform QPLEX at least marginally. These results demonstrate that Q+FIX succeeds in enhancing the performance of its fixees, raising them to a level comparable to QPLEX while maintaining a more stable convergence.

Table 1 shows the sizes of *mixing models* (for the all methods that have one). Notably, Q+FIX-{sum,lin} employ the smallest mixing models by a significant margin, indicating that their performance is a consequence of our pro-posed mixing structure rather than model size.

Appendix F.1 contains additional discussion on the SMACv2 evaluation, implementation de-

Table 1: SMACv2 mixer sizes (smallest highlighted).

|  | Protoss | | | Terran, Zerg | | |
|---|---|---|---|---|---|---|
|  | 5vs5 | 10vs10 | 20vs20 | 5vs5 | 10vs10 | 20vs20 |
| QMIX | 38 k | 83 k | 201 k | 36 k | 79 k | 194 k |
| QPLEX | 135 k | 326 k | 882 k | 126 k | 308 k | 846 k |
| Q+FIX-sum | 20 k | 50 k | 138 k | 19 k | 48 k | 133 k |
| Q+FIX-mono | 54 k | 180 k | 743 k | 50 k | 169 k | 708 k |
| Q+FIX-lin | 21 k | 51 k | 140 k | 19 k | 48 k | 135 k |

tails and chosen metrics, additional *winrate* results, *probability-of-improvement* Agarwal et al. (2021) results, and an evaluation of model sizes for Q+FIX-mono and QMIX.

**Overcooked** `JaxMARL` Rutherford et al. (2024) provides baseline implementations for Over-cooked Carroll et al. (2020), another popular benchmark for cooperative multi-agent control focused on throughput efficiency. Different layouts represent different challenges, e.g., subtask assignment and synchronization for efficiency. `JaxMARL` provides base implementations for independent Q-learning (IQL), VDN and QMIX (but not QPLEX), and we implemented Q+FIX-{sum,mono,lin}. Appendix F.2 contains further discussion on these tasks, and additional results.

Fig. 3 contains the evaluation results for three challenging layouts: `Coordination-Ring`, `Forced-Coordination`, and `Counter-Circuit`. In contrast to the SMACv2 results, this time it is specifically Q+FIX-mono to outperform other baselines and Q+FIX variants, indicating that there are concrete situations where Q+FIX is able to exploit a more complex fixee structure. Aside from this difference, these results reaffirm the ability of QFIX to greatly expand the represen-tation capabilities of the underlying fixees, enabling higher performances.

## 6 CONCLUSIONS

In this work, we have advanced our understanding of the IGM function class by proposing a simple formulation of the IGM property. From this formulation, we were able to naturally derive QFIX, a novel family of value function decomposition methods that enhance prior methods via a simple weighted transformation of their outputs, and allows the derivation and implementation of various IGM-complete models that are significantly simpler than QPLEX. Our empirical evaluation on mul-tiple SMACv2 and Overcooked tasks demonstrates that QFIX models succeed in (i) enhancing the performance of prior incomplete models like VDN and QMIX, (ii) achieving similar or better per-formance than QPLEX, with better convergence stability, and (iii) all this while requiring smaller mixing models. Our contribution not only represents a novel approach that performs well, but also opens the door for new methods based on the QFIX framework.

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

## A    IS QPLEX TRULY IGM-COMPLETE?

In this section, we take a closer look at (Wang et al., 2016, Proposition 2) which appeals to the universal approximation theorem (UAT) to claim that that QPLEX is IGM-complete. We will identify a technical issue that makes many "strong" forms of UAT not formally applicable, and come to the primary conclusions that (i) only "weak" forms of UAT are applicable to QPLEX, and (ii) consequently, QPLEX is able to approximate "only" the function class of *measurable* IGM values. To be clear, this is far from being a strict limitation in practice, as the class of measurable functions is extremely wide and contains any reasonable function to model, and mostly excludes deeply degenerate cases.

The main goal of this discussion is to be more specific in regards to *what* version of UAT is applicable to methods like QPLEX (and QFIX), and what kinds of convergence guarantees they actually entail.

Part of the issue at hand is that UATs come in a variety of forms, each making different assumptions on the model and establishing different notions of approximation to different classes of target functions. The UATs of Cybenko (1989) and of Pinkus (1999) are among the most well known, and are formulated in terms of *uniform convergence*, a strong notion of approximation that is only applicable to approximate *continuous* functions. However, other forms of UAT are applicable to approximate wider classes of functions, although they are also typically associated with weaker notions of approximation. Hornik (1991, Theorem 1) establishes a form of UAT that is applicable to functions in the Lebesgue spaces $L^p$ and entails *convergence in $p$-norm*. Hornik (1991) also informally formulates a corollary that is applicable to functions that are merely *measurable*, and "only" entails *convergence in measure $\mu$*.

**Which universal approximation theorem?** The appeal to UAT made by Wang et al. (2020) cites a form of UAT that is analogous to those of Cybenko (1989); Pinkus (1999) that are formally applicable to continuous functions only. However we note two relevant details: (i) QPLEX constructs $\hat{Q}_{\mathrm{PLEX}}$ by composing individual values via models $w_i$, $b_i$ and $\lambda_i$; therefore any appeal to UAT must refer to these models rather than $\hat{Q}_{\mathrm{PLEX}}$ as a whole. (ii) The proof of Proposition 2 is based on constructing a piece-wise target $\lambda_i^*(\boldsymbol{h}, \boldsymbol{a})$ that is clearly *not* guaranteed to be continuous. These are

$$\lambda_i^*(\boldsymbol{h}, \boldsymbol{a}) = \begin{cases} \frac{1}{N} \frac{A(\boldsymbol{h}, \boldsymbol{a})}{A_i(h_i, a_i)} & \text{when } A_i(h_i, a_i) < 0\,, \\ \text{any value} & \text{when } A_i(h_i, a_i) = 0\,, \end{cases} \tag{21}$$

where $A(\boldsymbol{h}, \boldsymbol{a})$ is the advantage of the target IGM value function. Clearly, as the target $\lambda_i^*$ is not continuous, it is improper to appeal to a form of UAT that is based on continuous targets.

**Resolution** To resolve this technicality, we must find a version of UAT that is applicable to a target like $\lambda_i^*$. It is not immediately clear that $\lambda_i^*$ belongs to a Lebesgue space $L^p$, or what kinds of *simple* assumptions can be formulated to make it so. As a simple resolution, we instead appeal to the weaker form of UAT by Cybenko (1989) (presented informally in the discussion section) based on *measurable* functions. However, even this form of UAT still requires some technical assumptions.

To guarantee that $\lambda_i^*$ is measurable, it is sufficient to assume that $Q_i(h_i, a_i)$ and $Q(\boldsymbol{h}, \boldsymbol{a})$ are measurable functions. Then,

- $V_i(h_i)$, $V(\boldsymbol{h})$, $A_i(h_i, a_i)$, $A(\boldsymbol{h}, \boldsymbol{a})$ are measurable;
- $\mathrm{argmax}_{h_i, a_i} A_i(h_i, a_i)$ (the preimage of $A_i(h_i, a_i) = 0$) is a measurable set;
- $\lambda_i^*$ is a piece-wise function defined by combining measurable functions partitioned in (two) measurable sets, and is therefore also measurable.

This is sufficient to guarantee convergence to $\lambda_i^*$ in measure. Technically this assumption means that there are *non-measurable* IGM values that cannot be approximated by QPLEX (nor QFIX). However, we reiterate that this is not a practical concern as (i) they represent an insignificant subset of all IGM values, and (ii) they are degenerate and unlikely to match realistic and desirable notions of values.

# B   PROOFS

## B.1   PROOF OF PROPOSITION 3

We prove the two statements separately.

### $Q_{\mathrm{IGM}}$ satisfies IGM

*Proof.* For any given joint history $\boldsymbol{h}$, let $a_i^* \in \mathrm{argmax}_{a_i} Q_i(h_i, a_i)$ denote any maximal action according to the individual utilities, and let $\boldsymbol{a}^* = (a_1^*, \ldots, a_N^*)$ be a joint action constructed accordingly. For any $\boldsymbol{a}^*$ constructed this way, the corresponding advantage utilities are zero $\forall i\, (u_i^* = 0)$, and

$$Q_{\mathrm{IGM}}(\boldsymbol{h}, \boldsymbol{a}^*) = w(\boldsymbol{h}, \boldsymbol{a}^*) \underbrace{f(u_1^*, \ldots, u_N^*)}_{=0} + b(\boldsymbol{h})$$

$$= b(\boldsymbol{h})\,. \tag{22}$$

For any other $\boldsymbol{a}$, we have at least one strictly negative utility $\exists i\, (u_i < 0)$, and

$$Q_{\mathrm{IGM}}(\boldsymbol{h}, \boldsymbol{a}) = \underbrace{w(\boldsymbol{h}, \boldsymbol{a})}_{>0} \underbrace{f(u_1, \ldots, u_N)}_{<0} + b(\boldsymbol{h})$$

$$< b(\boldsymbol{h})\,. \tag{23}$$

Therefore $\boldsymbol{a}^* \in \mathrm{argmax}_{\boldsymbol{a}} Q_{\mathrm{IGM}}(\boldsymbol{h}, \boldsymbol{a})$, and the actions that maximize the individual utilities also maximize the joint value.

$\square$

### $Q_{\text{IGM}}$ is IGM-complete

*Proof by mutual inclusion.* Let us denote the function class of $Q_{\text{IGM}}$ as $\mathcal{FC}(Q_{\text{IGM}})$, and the IGM-complete function class as $\mathcal{FC}_{\text{IGM}}$. We prove $\mathcal{FC}(Q_{\text{IGM}}) = \mathcal{FC}_{\text{IGM}}$ by mutual inclusion:

1. $Q \in \mathcal{FC}(Q_{\text{IGM}}) \implies Q \in \mathcal{FC}_{\text{IGM}}$, i.e., $Q_{\text{IGM}}$ satisfies IGM (already proven above),

2. $Q \in \mathcal{FC}_{\text{IGM}} \implies Q \in \mathcal{FC}(Q_{\text{IGM}})$, i.e., any IGM function is representable by $Q_{\text{IGM}}$.

Step 1 was already proven earlier. Next, we prove step 2.

Let $Q_i(h_i, a_i)$ and $Q(\boldsymbol{h}, \boldsymbol{a})$ denote an arbitrary set of individual and joint values that satisfy IGM, i.e., $Q \in \mathcal{FC}_{\text{IGM}}$. Let us denote the usual corresponding individual values and advantages as follows,

$$V_i(h_i) = \max_{a_i} Q_i(h_i, a_i)\,, \qquad\qquad A_i(h_i, a_i) = Q_i(h_i, a_i) - V_i(h_i)\,, \qquad (24)$$

$$V(\boldsymbol{h}) = \max_{\boldsymbol{a}} Q(\boldsymbol{h}, \boldsymbol{a})\,, \qquad\qquad A(\boldsymbol{h}, \boldsymbol{a}) = Q(\boldsymbol{h}, \boldsymbol{a}) - V(\boldsymbol{h})\,, \qquad (25)$$

with the usual shorthand $q_i = Q_i(h_i, a_i)$ and $v_i = V_i(h_i)$, and $u_i = A_i(h_i, a_i)$.

For any $f$ that satisfies the requirements of Eq. (8), let $w$ and $b$ be defined as follows,

$$b(\boldsymbol{h}) = V(\boldsymbol{h})\,, \qquad (26)$$

$$w(\boldsymbol{h}, \boldsymbol{a}) = \begin{cases} \frac{A(\boldsymbol{h}, \boldsymbol{a})}{f(u_1, \ldots, u_N)}\,, & \text{if } f(u_1, \ldots, u_N) \neq 0\,, \\ \text{any value}\,, & \text{otherwise}\,. \end{cases} \qquad (27)$$

For any given joint history $\boldsymbol{h}$, let $a_i^* \in \operatorname{argmax}_{a_i} Q_i(h_i, a_i)$ denote a maximal action according to the individual utilities, and $\boldsymbol{a}^* = (a_1^*, \ldots, a_N^*)$ the corresponding joint action. Given that $Q$ satisfies IGM by assumption, we have $\boldsymbol{a}^* \in \operatorname{argmax}_{\boldsymbol{a}} Q(\boldsymbol{h}, \boldsymbol{a})$, and $Q(\boldsymbol{h}, \boldsymbol{a}^*) = \max_{\boldsymbol{a}} Q(\boldsymbol{h}, \boldsymbol{a}) = V(\boldsymbol{h})$.

For any $\boldsymbol{a}^*$ constructed this way, the corresponding advantage utilities are zero $\forall i \, (u_i = 0)$, and

$$\begin{aligned} Q_{\text{IGM}}(\boldsymbol{h}, \boldsymbol{a}^*) &= w(\boldsymbol{h}, \boldsymbol{a}^*) f(u_1, \ldots, u_N) + b(\boldsymbol{h}) \\ &= w(\boldsymbol{h}, \boldsymbol{a}^*) \underbrace{f(0, \ldots, 0)}_{=0} + b(\boldsymbol{h}) \\ &= V(\boldsymbol{h}) \\ &= Q(\boldsymbol{h}, \boldsymbol{a}^*)\,. \end{aligned} \qquad (28)$$

For any other $\boldsymbol{a}$, we have at least one strictly negative utility $\exists i \, (u_i < 0)$, and

$$\begin{aligned} Q_{\text{IGM}}(\boldsymbol{h}, \boldsymbol{a}) &= w(\boldsymbol{h}, \boldsymbol{a}) f(u_1, \ldots, u_N) + b(\boldsymbol{h}) \\ &= \frac{A(\boldsymbol{h}, \boldsymbol{a})}{f(u_1, \ldots, u_N)} f(u_1, \ldots, u_N) + V(\boldsymbol{h}) \\ &= A(\boldsymbol{h}, \boldsymbol{a}) + V(\boldsymbol{h}) \\ &= Q(\boldsymbol{h}, \boldsymbol{a})\,. \end{aligned} \qquad (29)$$

In either case, $Q_{\text{IGM}}(\boldsymbol{h}, \boldsymbol{a}) = Q(\boldsymbol{h}, \boldsymbol{a})$ for all inputs. Therefore $Q \in \mathcal{FC}_{\text{IGM}} \implies Q \in \mathcal{FC}(Q_{\text{IGM}})$.

$\square$

### B.2 Proof of Proposition 4

*Proof.* Equation (11) satisfies the form and requirements of Eq. (8). Therefore, IGM follows from Proposition 3. Assuming target IGM values that are measurable, then the targets constructed in the proof of Proposition 3 are also measurable, and we can appeal to the universal approximation theorems of Hornik (1991) to show that $w, b$ are able to approximate such targets. (also see Appendix A for a similar discussion relating to QPLEX). $\square$

### B.3 PROOF OF PROPOSITION 5

*Proof.* QFIX-lin is a monotonic function of individual advantages and therefore satisfies IGM. QFIX-lin is also a generalization of QFIX-sum, therefore its function class is a superset of the QFIX-sum function class, i.e., the class of measurable IGM values. Therefore, QFIX-lin can represent all measurable functions that satisfy IGM, and none of those that do not satisfy IGM. □

## C DERIVATIONS

This section contains explicit long-form derivations that had to be removed from the main document due to space limitations.

### C.1 VDN MAXIMAL VALUES AND ADVANTAGES

As a reminder, VDN action-values are defined as $\hat{Q}_{\text{VDN}}(\boldsymbol{h}, \boldsymbol{a}) \doteq \sum_i \hat{Q}_i(h_i, a_i)$. Due to the the linear (monotonic) mixing structure, the joint maximal values $\hat{V}_{\text{VDN}}(\boldsymbol{h})$ can be expressed as the sum of the individual maximal values,

$$
\begin{aligned}
\hat{V}_{\text{VDN}}(\boldsymbol{h}) &\doteq \max_{\boldsymbol{a}} \hat{Q}_{\text{VDN}}(\boldsymbol{h}, \boldsymbol{a}) \\
&= \max_{a_1, \ldots, a_N} \sum_i \hat{Q}_i(h_i, a_i) \\
&= \sum_i \max_{a_i} \hat{Q}_i(h_i, a_i) \qquad \text{(monotonicity)} \\
&= \sum_i \hat{V}_i(h_i) \,,
\end{aligned}
\tag{30}
$$

and the joint advantages $\hat{A}_{\text{VDN}}(\boldsymbol{h}, \boldsymbol{a})$ can be expressed as the sum of the individual advantages,

$$
\begin{aligned}
\hat{A}_{\text{VDN}}(\boldsymbol{h}, \boldsymbol{a}) &\doteq \hat{Q}_{\text{VDN}}(\boldsymbol{h}, \boldsymbol{a}) - \hat{V}_{\text{VDN}}(\boldsymbol{h}) \\
&= \sum_i \hat{Q}_i(h_i, a_i) - \sum_i \hat{V}_i(h_i) \\
&= \sum_i \hat{Q}_i(h_i, a_i) - \hat{V}_i(h_i) \\
&= \sum_i \hat{A}_i(h_i, a_i) \,.
\end{aligned}
\tag{31}
$$

### C.2 QMIX MAXIMAL VALUES AND ADVANTAGES

As a reminder, QMIX action-values are defined as $\hat{Q}_{\text{MIX}}(\boldsymbol{h}, \boldsymbol{a}) \doteq f_{\text{mono}}(q_1, \ldots, q_N)$. Due to the monotonic mixing structure, the joint maximal values $\hat{V}_{\text{MIX}}(\boldsymbol{h})$ can be expressed as the monotonic mixing of the individual maximal values,

$$
\begin{aligned}
\hat{V}_{\text{MIX}}(\boldsymbol{h}) &\doteq \max_{\boldsymbol{a}} \hat{Q}_{\text{MIX}}(\boldsymbol{h}, \boldsymbol{a}) \\
&= \max_{a_1, \ldots, a_N} f_{\text{mono}}\left(\hat{Q}_1(h_1, a_1), \ldots, \hat{Q}_N(h_N, a_N)\right) \\
&= f_{\text{mono}}\left(\max_{a_1} \hat{Q}_1(h_1, a_1), \ldots, \max_{a_N} \hat{Q}_N(h_N, a_N)\right) \qquad \text{(monotonicity)} \\
&= f_{\text{mono}}\left(\hat{V}_1(h_1), \ldots, \hat{V}_N(h_N)\right) \\
&= f_{\text{mono}}(v_1, \ldots, v_N) \,,
\end{aligned}
\tag{32}
$$

and the joint advantages $\hat{A}_{\text{MIX}}(\boldsymbol{h}, \boldsymbol{a})$ can be expressed as the corresponding difference,

$$
\begin{aligned}
\hat{A}_{\text{MIX}}(\boldsymbol{h}, \boldsymbol{a}) &\doteq \hat{Q}_{\text{MIX}}(\boldsymbol{h}, \boldsymbol{a}) - \hat{V}_{\text{MIX}}(\boldsymbol{h}) \\
&= f_{\text{mono}}(q_1, \ldots, q_N) - f_{\text{mono}}(v_1, \ldots, v_N) \,.
\end{aligned}
\tag{33}
$$

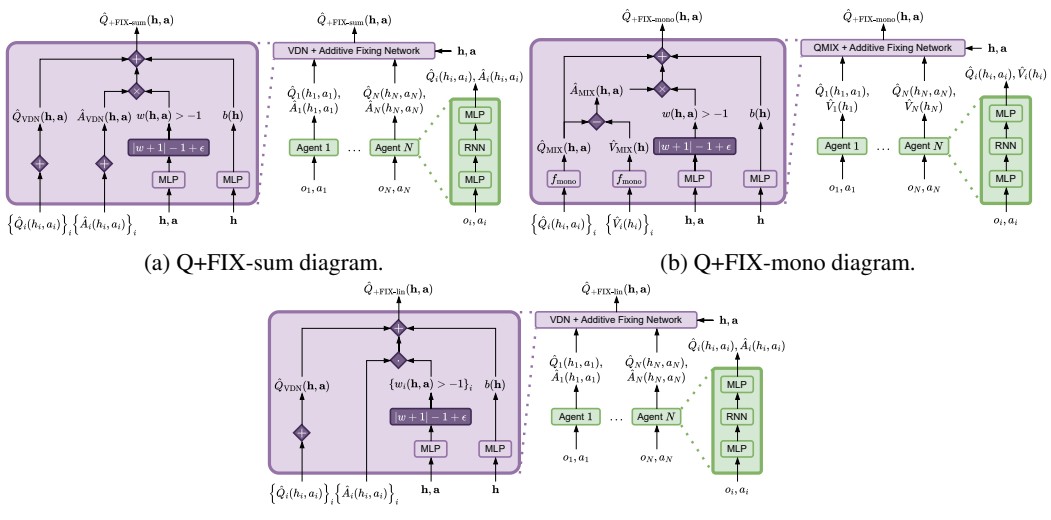

(a) Q+FIX-sum diagram.
(b) Q+FIX-mono diagram.

(c) Q+FIX-lin diagram.

Figure 4: Specialized diagrams for Q+FIX-sum, Q+FIX-mono, and Q+FIX-lin.

### C.3 QFIX-SUM

QFIX-sum is an instance of QFIX based on VDN as fixee model, $\hat{Q}_{\text{fixee}}(\boldsymbol{h}, \boldsymbol{a}) = \hat{Q}_{\text{VDN}}(\boldsymbol{h}, \boldsymbol{a})$. From Eq. (31), we have that the VDN joint advantage is given as the sum of individual advantages (hence the "-sum" suffix). Therefore, QFIX-sum is simply obtained as

$$\hat{Q}_{\text{FIX-sum}}(\boldsymbol{h}, \boldsymbol{a}) \doteq w(\boldsymbol{h}, \boldsymbol{a})\hat{A}_{\text{VDN}}(\boldsymbol{h}, \boldsymbol{a}) + b(\boldsymbol{h})$$
$$= w(\boldsymbol{h}, \boldsymbol{a}) \sum_i \hat{A}_i(h_i, a_i) + b(\boldsymbol{h}) \,. \tag{34}$$

### C.4 QFIX-MONO

QFIX-mono is an instance of QFIX based on QMIX as fixee model, $\hat{Q}_{\text{fixee}}(\boldsymbol{h}, \boldsymbol{a}) = \hat{Q}_{\text{MIX}}(\boldsymbol{h}, \boldsymbol{a})$. From Eq. (33), we have that the QMIX advantage is given as a difference between monotonic compositions of individual utilities (hence the "-mono" suffix). Therefore, QFIX-mono is simply obtained as

$$\hat{Q}_{\text{FIX-mono}}(\boldsymbol{h}, \boldsymbol{a}) \doteq w(\boldsymbol{h}, \boldsymbol{a})\hat{A}_{\text{MIX}}(\boldsymbol{h}, \boldsymbol{a}) + b(\boldsymbol{h})$$
$$= w(\boldsymbol{h}, \boldsymbol{a})(f_{\text{mono}}(q_1, \ldots, q_N) - f_{\text{mono}}(v_1, \ldots, v_N)) + b(\boldsymbol{h}) \,. \tag{35}$$

### C.5 Q+FIX-SUM

Q+FIX-sum is an instance of Q+FIX based on VDN as fixee model, $\hat{Q}_{\text{fixee}}(\boldsymbol{h}, \boldsymbol{a}) = \hat{Q}_{\text{VDN}}(\boldsymbol{h}, \boldsymbol{a})$ and $\hat{A}_{\text{fixee}}(\boldsymbol{h}, \boldsymbol{a}) = \hat{A}_{\text{VDN}}(\boldsymbol{h}, \boldsymbol{a})$, also equivalent to the additive formulation of QFIX-sum. Therefore, Q+FIX-sum is simply obtained as

$$\hat{Q}_{\text{+FIX-sum}} \doteq \hat{Q}_{\text{VDN}}(\boldsymbol{h}, \boldsymbol{a}) + w(\boldsymbol{h}, \boldsymbol{a})\hat{A}_{\text{VDN}}(\boldsymbol{h}, \boldsymbol{a}) + b(\boldsymbol{h})$$
$$\doteq \sum_i \hat{Q}_i(\boldsymbol{h}, \boldsymbol{a}) + w(\boldsymbol{h}, \boldsymbol{a}) \sum_i \hat{A}_i(\boldsymbol{h}, \boldsymbol{a}) + b(\boldsymbol{h}) \,. \tag{36}$$

Figure 4a shows a graphical diagram for Q+FIX-sum.

### C.6 Q+FIX-MONO

Q+FIX-mono is an instance of Q+FIX based on QMIX as fixee model, $\hat{Q}_{\text{fixee}}(\boldsymbol{h}, \boldsymbol{a}) = \hat{Q}_{\text{MIX}}(\boldsymbol{h}, \boldsymbol{a})$ and $\hat{A}_{\text{fixee}}(\boldsymbol{h}, \boldsymbol{a}) = \hat{A}_{\text{MIX}}(\boldsymbol{h}, \boldsymbol{a})$, also equivalent to the additive formulation of QFIX-mono. There-

fore, Q+FIX-mono is simply obtained as

$$
\begin{aligned}
\hat{Q}_{+\text{FIX-mono}} &\doteq \hat{Q}_{\text{VDN}}(\boldsymbol{h}, \boldsymbol{a}) + w(\boldsymbol{h}, \boldsymbol{a})\hat{A}_{\text{VDN}}(\boldsymbol{h}, \boldsymbol{a}) + b(\boldsymbol{h}) \\
&\doteq f_{\text{mono}}(q_1, \ldots, q_N) + w(\boldsymbol{h}, \boldsymbol{a})\left(f_{\text{mono}}(q_1, \ldots, q_N) - f_{\text{mono}}(v_1, \ldots, v_N)\right) + b(\boldsymbol{h}) .
\end{aligned}
\tag{37}
$$

Figure 4b shows a graphical diagram for Q+FIX-mono.

## C.7 Q+FIX-LIN

Q+FIX-lin is the additive formulation of QFIX-lin. Just as QFIX-lin is not formally a member of the QFIX family, but rather a generalization of QFIX-sum, so is Q+FIX-lin not formally a member of Q+FIX, but rather a generalization of Q+FIX-sum. Given that QFIX-lin is obtained by introducing per-agent weights $w_i(\boldsymbol{h}, \boldsymbol{a})$, Q+FIX-lin is simply obtained as

$$
\hat{Q}_{+\text{FIX-lin}} \doteq \sum_i \hat{Q}_i(h_i, a_i) + \sum_i w_i(\boldsymbol{h}, \boldsymbol{a})\hat{A}_i(h_i, a_i) + b(\boldsymbol{h}) .
$$

Figure 4c shows a graphical diagram for Q+FIX-lin.

## D WHY DOES DETACHING THE ADVANTAGES HELP Q+FIX?

First, we note that the gradients $\nabla_{\theta_i}\hat{Q}_{+\text{FIX}}(\boldsymbol{h}, \boldsymbol{a})$ when the advantages *are not* detached are

$$
\begin{aligned}
\nabla_{\theta_i}\hat{Q}_{+\text{FIX}}(\boldsymbol{h}, \boldsymbol{a}) &= \nabla_{\theta_i}\hat{Q}_{\text{fixee}}(\boldsymbol{h}, \boldsymbol{a}) + w(\boldsymbol{h}, \boldsymbol{a})\nabla_{\theta_i}\hat{A}_{\text{fixee}}(\boldsymbol{h}, \boldsymbol{a}) \\
&= \nabla_{\theta_i}\hat{V}_{\text{fixee}}(\boldsymbol{h}) + (w(\boldsymbol{h}, \boldsymbol{a}) + 1)\nabla_{\theta_i}\hat{A}_{\text{fixee}}(\boldsymbol{h}, \boldsymbol{a}) .
\end{aligned}
\tag{38}
$$

It seems plausible that there may be values of $w(\boldsymbol{h}, \boldsymbol{a})$ that could result in non-ideal gradient signals. For example, a low fixing weight $w(\boldsymbol{h}, \boldsymbol{a}) \approx -1$ results in a dampened gradient $\nabla_{\theta_i}\hat{Q}_{+\text{FIX}}(\boldsymbol{h}, \boldsymbol{a}) \approx \nabla_{\theta_i}\hat{V}_{\text{fixee}}(\boldsymbol{h})$, that is notably independent on actions. On the other end of the spectrum, a very large fixing weight $w(\boldsymbol{h}, \boldsymbol{a}) \gg -1$ results in a gradient that is dominated by the highly-weighted advantage component, overcoming the value component, $\nabla_{\theta_i}\hat{Q}_{+\text{FIX}}(\boldsymbol{h}, \boldsymbol{a}) \approx w(\boldsymbol{h}, \boldsymbol{a})\nabla_{\theta_i}\hat{A}_{\text{fixee}}(\boldsymbol{h}, \boldsymbol{a})$. On each end of the spectrum, the gradient will propagate almost exclusively through the values $\nabla_{\theta_i}\hat{V}_{\text{fixee}}(\boldsymbol{h})$ or through the advantages $\nabla_{\theta_i}\hat{A}_{\text{fixee}}(\boldsymbol{h}, \boldsymbol{a})$.

On the other hand, the gradients $\nabla_{\theta_i}\hat{Q}_{+\text{FIX}}(\boldsymbol{h}, \boldsymbol{a})$ when the advantages *are* detached are

$$
\begin{aligned}
\nabla_{\theta_i}\hat{Q}_{+\text{FIX}}(\boldsymbol{h}, \boldsymbol{a}) &= \nabla_{\theta_i}\hat{Q}_{\text{fixee}}(\boldsymbol{h}, \boldsymbol{a}) \\
&= \nabla_{\theta_i}\hat{V}_{\text{fixee}}(\boldsymbol{h}) + \nabla_{\theta_i}\hat{A}_{\text{fixee}}(\boldsymbol{h}, \boldsymbol{a}) ,
\end{aligned}
\tag{39}
$$

and are invariant to the fixing structure, equally dependent on the value and advantage components.

## E STATE-BASED QFIX

In this section, we extend some of the theory of QFIX to the state-based case. As mentioned in the main document, we consider two cases of state-based QFIX, a *history-state* case and *state-only* case, which differ in what information is provided to the fixing network. The derivations and proofs will follow closely those of the stateless case, although not all conclusions will transfer to all state-based cases. Primarily, we will find that state-only QFIX (like other state-only variants of other methods) is not able to represent the full IGM-complete space of value functions.

### E.1 HISTORY-STATE QFIX

Consider a history-state variant of $Q_{\text{IGM}}$ from Eq. (8) defined as follows,

$$
Q_{\text{IGM}}(\boldsymbol{h}, s, \boldsymbol{a}) \doteq w(\boldsymbol{h}, s, \boldsymbol{a})f(u_1, \ldots, u_N) + b(\boldsymbol{h}, s) ,
\tag{40}
$$

where $u_i$ and $f$ are defined as in Section 4.1, $w \colon \mathcal{H} \times \mathcal{S} \times \mathcal{A} \to \mathbb{R}_{>0}$ is an arbitrary positive function of joint history, state, and joint action, $b \colon \mathcal{H} \times \mathcal{S} \to \mathbb{R}$ is an arbitrary function of joint history and state. As in the stateless case, $Q_{\text{IGM}}(\boldsymbol{h}, s, \boldsymbol{a})$ denotes a relationship where any deviation from individual maximality is transformed into an arbitrary deviation from joint maximality.

**Proposition 6.** *For any f, w, and b, values $\{Q_i\}_{i \in \mathcal{I}}$ and $Q_{\text{IGM}}$ satisfy state-based IGM.*

*Proof.* This proof follows the same structure as that for Proposition 3.

For any given joint history $\boldsymbol{h}$, let $a_i^* = \text{argmax}_{a_i} Q_i(h_i, a_i)$ denote the maximal action according to the individual utilities, and $\boldsymbol{a}^* = (a_i^*, \ldots, a_N^*)$ the joint action constructed by those individual actions. We prove that $Q_{\text{IGM}}$ satisfies state-based IGM in two steps:

1. $\boldsymbol{a}^* = \text{argmax}_{\boldsymbol{a}} Q_{\text{IGM}}(\boldsymbol{h}, s, \boldsymbol{a})$, i.e., the individual maximal actions also maximize the joint history-state values.

2. $\boldsymbol{a}^* = \text{argmax}_{\boldsymbol{a}} \mathbb{E}_{s|\boldsymbol{h}} [Q_{\text{IGM}}(\boldsymbol{h}, s, \boldsymbol{a})]$, i.e., the individual maximal actions also maximize the marginalized joint history-state values.

**Step 1.** The advantage utilities corresponding to $\boldsymbol{a}^*$ are zero $\forall i(u_i = 0)$ by definition, and

$$Q_{\text{IGM}}(\boldsymbol{h}, s, \boldsymbol{a}^*) = w(\boldsymbol{h}, s, \boldsymbol{a}^*) \underbrace{f(u_1, \ldots, u_N)}_{=0} + b(\boldsymbol{h}, s)$$

$$= b(\boldsymbol{h}, s) \,. \tag{41}$$

For any other non-maximal action $\boldsymbol{a}$, we have at least one strictly negative utility $\exists i(u_i < 0)$, and

$$Q_{\text{IGM}}(\boldsymbol{h}, s, \boldsymbol{a}^*) = \underbrace{w(\boldsymbol{h}, s, \boldsymbol{a}^*)}_{>0} \underbrace{f(u_1, \ldots, u_N)}_{<0} + b(\boldsymbol{h}, s)$$

$$< b(\boldsymbol{h}, s) \,. \tag{42}$$

Therefore, $\boldsymbol{a}^* = \text{argmax}_{\boldsymbol{a}} Q_{\text{IGM}}(\boldsymbol{h}, s, \boldsymbol{a})$, and the actions that maximize the individual utilities also maximize the joint history-state value.

**Step 2.** Note that $\boldsymbol{a}^* = \text{argmax}_{\boldsymbol{a}} Q_{\text{IGM}}(\boldsymbol{h}, s, \boldsymbol{a})$ is valid for any state, at the very least because $\boldsymbol{a}^*$ are defined via the stateless individual utilities.

If $\boldsymbol{a}^*$ maximizes the joint history-state values for any given state, then it also maximizes the joint history-state values when marginalized over any distribution of state $p \in \Delta\mathcal{S}$, and $\boldsymbol{a}^* = \text{argmax}_{\boldsymbol{a}} \mathbb{E}_{s \sim p} [Q_{\text{IGM}}(\boldsymbol{h}, s, \boldsymbol{a})]$. This must be true also for the specific distribution $p(s) \doteq \Pr(s \mid \boldsymbol{h})$, and $\boldsymbol{a}^* = \text{argmax}_{\boldsymbol{a}} \mathbb{E}_{s|\boldsymbol{h}} [Q_{\text{IGM}}(\boldsymbol{h}, s, \boldsymbol{a})]$.

Therefore, the same actions $\boldsymbol{a}^*$ that maximize the individual utilities, also maximize the marginalized joint history-state values, satisfying the definition of state-based IGM in Definition 3.

$\square$

When it comes to a state-based form of IGM-complete function class, we must be very clear as to what it is that we are able to prove. We are not able to prove that $Q_{\text{IGM}}(\boldsymbol{h}, s, \boldsymbol{a})$ covers the whole state-based IGM function class of values that satisfy state-based IGM (we do not believe this is possible, though we will not go into that amount of detail here). Instead, we prove that the projected space of *stateless* values obtained by marginalizing the state-based values via $\mathbb{E}_{s|\boldsymbol{h}} [Q_{\text{IGM}}(\boldsymbol{h}, s, \boldsymbol{a})]$ is the IGM-complete function class.

**Proposition 7.** *For any f, and given free choice of w and b, the function class of $\{Q_i\}_{i \in \mathcal{I}}$ and projected $\mathbb{E}_{s|h} [Q_{\text{IGM}}]$ is IGM-complete.*

*Proof.* This proof follows the same structure as that for Proposition 3, although we consider the projected space stateless values $\mathbb{E}_{s|\boldsymbol{h}} [Q_{\text{IGM}}(\boldsymbol{h}, s, \boldsymbol{a})]$ obtained from the state-based values $Q_{\text{IGM}}(\boldsymbol{h}, s, \boldsymbol{a})$.

Let us denote the projected function class of $Q_{\text{IGM}}$ as $\mathcal{FC}(Q_{\text{IGM}})$, and the state-based IGM-complete function class as $\mathcal{FC}_{\text{IGM}}$. We prove the equivalence $\mathcal{FC}(Q_{\text{IGM}}) = \mathcal{FC}_{\text{IGM}}$ in two steps:

**Step 1.** $Q \in \mathcal{FC}(Q_{\text{IGM}}) \implies Q \in \mathcal{FC}_{\text{IGM}}$ follows directly from Proposition 6.

**Step 2.** Let $Q_i(h_i, a_i)$ and $Q(\boldsymbol{h}, \boldsymbol{a})$ denote an arbitrary set of individual and joint values that satisfy IGM, i.e., $Q \in \mathcal{FC}_{\text{IGM}}$. Let us denote the usual corresponding values and advantages as follows,

$$V_i(h_i) = \max_{a_i} Q_i(h_i, a_i), \qquad\qquad A_i(h_i, a_i) = Q_i(h_i, a_i) - V_i(h_i), \qquad (43)$$

*but*, let us define a different notion of joint values and advantages for this history-state case (note the stateless $V$, state-based $A$),

$$V(\boldsymbol{h}) = \max_{\boldsymbol{a}} Q(\boldsymbol{h}, \boldsymbol{a}), \qquad\qquad A(\boldsymbol{h}, \boldsymbol{a}) = Q(\boldsymbol{h}, \boldsymbol{a}) - V(\boldsymbol{h}), \qquad (44)$$

with the usual shorthand $q_i = Q_i(h_i, a_i)$ and $v_i = V_i(h_i)$, and $u_i = A_i(h_i, a_i)$.

For any $f$ that satisfies the requirements of Eq. (41), let $w$ and $b$ be defined as follows,

$$b(\boldsymbol{h}, s) = V(\boldsymbol{h}), \qquad (45)$$

$$w(\boldsymbol{h}, s, \boldsymbol{a}) = \begin{cases} \frac{A(\boldsymbol{h}, \boldsymbol{a})}{f(u_1, \ldots, u_N)}, & \text{if } f(u_1, \ldots, u_N) \neq 0, \\ \text{any value}, & \text{otherwise}. \end{cases} \qquad (46)$$

These definitions effectively create state-based values $Q_{\text{IGM}}(\boldsymbol{h}, s, \boldsymbol{a})$ that are state-independent, and functionally equivalent to stateless values $Q_{\text{IGM}}(\boldsymbol{h}, \boldsymbol{a})$. Although this appears to be a severe misuse of the additional state information, it is sufficient to prove the claim that the projected space of stateless values obtained via marginalization $\mathbb{E}_{s|\boldsymbol{h}}[Q_{\text{IGM}}(\boldsymbol{h}, \boldsymbol{a})]$ is IGM-complete. It's easy to see that the rest of the proof can not proceed as in Proposition 3.

For any given joint history $\boldsymbol{h}$, let $a_i^* = \operatorname{argmax}_{a_i} Q_i(h_i, a_i)$ denote the maximal action according to the individual utilities, and $\boldsymbol{a}^* = (a_1^*, \ldots, a_N^*)$ the corresponding joint action. Given that $Q$ satisfies IGM by assumption, we have $\boldsymbol{a}^* = \operatorname{argmax}_{\boldsymbol{a}} Q(\boldsymbol{h}, \boldsymbol{a})$, and $Q(\boldsymbol{h}, \boldsymbol{a}^*) = \max_{\boldsymbol{a}} Q(\boldsymbol{h}, \boldsymbol{a}) = V(\boldsymbol{h})$.

For this joint action $\boldsymbol{a}^*$, the corresponding individual advantage utilities are zero $\forall i \, (u_i = 0)$ by definition, and

$$\begin{aligned} Q_{\text{IGM}}(\boldsymbol{h}, s, \boldsymbol{a}^*) &= w(\boldsymbol{h}, s, \boldsymbol{a}^*) f(u_1, \ldots, u_N) + b(\boldsymbol{h}, s) \\ &= w(\boldsymbol{h}, s, \boldsymbol{a}^*) \underbrace{f(0, \ldots, 0)}_{=0} + b(\boldsymbol{h}, s) \\ &= V(\boldsymbol{h}) \\ &= Q(\boldsymbol{h}, \boldsymbol{a}^*). \end{aligned} \qquad (47)$$

For any other non-maximal action $\boldsymbol{a}^\dagger$, we have at least one strictly negative utility $\exists i \, (u_i < 0)$, and

$$\begin{aligned} Q_{\text{IGM}}(\boldsymbol{h}, s, \boldsymbol{a}^\dagger) &= w(\boldsymbol{h}, s, \boldsymbol{a}^\dagger) f(u_1, \ldots, u_N) + b(\boldsymbol{h}, s) \\ &= \frac{A(\boldsymbol{h}, \boldsymbol{a}^\dagger)}{f(u_1, \ldots, u_N)} f(u_1, \ldots, u_N) + V(\boldsymbol{h}) \\ &= A(\boldsymbol{h}, \boldsymbol{a}^\dagger) + V(\boldsymbol{h}) \\ &= Q(\boldsymbol{h}, \boldsymbol{a}^\dagger). \end{aligned} \qquad (48)$$

In either case, $Q_{\text{IGM}}(\boldsymbol{h}, s, \boldsymbol{a}) = Q(\boldsymbol{h}, \boldsymbol{a})$ for all joint histories, states, and actions, which trivially implies $\mathbb{E}_{s|\boldsymbol{h}}[Q_{\text{IGM}}(\boldsymbol{h}, s, \boldsymbol{a})] = Q(\boldsymbol{h}, \boldsymbol{a})$. Therefore $Q \in \mathcal{FC}_{\text{IGM}} \implies Q \in \mathcal{FC}(Q_{\text{IGM}})$.

$\square$

### E.2 State-only QFIX

Consider a state-only variant of $Q_{\text{IGM}}$ from Eq. (8) defined as follows,

$$Q_{\text{IGM}}(\boldsymbol{h}, s, \boldsymbol{a}) \doteq w(s, \boldsymbol{a}) f(u_1, \ldots, u_N) + b(s), \qquad (49)$$

where $u_i$ and $f$ are defined as in Section 4.1, $w \colon \mathcal{S} \times \boldsymbol{\mathcal{A}} \to \mathbb{R}_{>0}$ is an arbitrary positive function of joint history, state, and joint action, $b \colon \mathcal{S} \to \mathbb{R}$ is an arbitrary function of joint history and state. As in the stateless case, $Q_{\text{IGM}}(\boldsymbol{h}, s, \boldsymbol{a})$ denotes a relationship where any deviation from individual maximality is transformed into an arbitrary deviation from joint maximality. Note that the name *state-only* refers moreso to the fixing models $w, b$, than the values as a whole that remain at least in part history-based due to the dependence on the individual history-based utilities.

**Proposition 8.** *For any $f$, $w$, and $b$, values $\{Q_i\}_{i \in \mathcal{I}}$ and $Q_{\text{IGM}}$ satisfy state-based IGM.*

*Proof.* This proof follows the same structure as that for Proposition 3.

For any given joint history $\boldsymbol{h}$, let $a_i^* = \text{argmax}_{a_i} Q_i(h_i, a_i)$ denote the maximal action according to the individual utilities, and $\boldsymbol{a}^* = (a_i^*, \ldots, a_N^*)$ the joint action constructed by those individual actions. We prove that $Q_{\text{IGM}}$ satisfies state-based IGM in two steps:

1. $\boldsymbol{a}^* = \text{argmax}_{\boldsymbol{a}} Q_{\text{IGM}}(\boldsymbol{h}, s, \boldsymbol{a})$, i.e., the individual maximal actions also maximize the state-only values.

2. $\boldsymbol{a}^* = \text{argmax}_{\boldsymbol{a}} \mathbb{E}_{s|\boldsymbol{h}} [Q_{\text{IGM}}(\boldsymbol{h}, s, \boldsymbol{a})]$, i.e., the individual maximal actions also maximize the marginalized joint state-only values.

**Step 1.** The advantage utilities corresponding to $\boldsymbol{a}^*$ are zero $\forall i (u_i = 0)$ by definition, and

$$Q_{\text{IGM}}(\boldsymbol{h}, s, \boldsymbol{a}^*) = w(s, \boldsymbol{a}^*) \underbrace{f(u_1, \ldots, u_N)}_{=0} + b(s)$$
$$= b(s). \tag{50}$$

For any other non-maximal action $\boldsymbol{a}$, we have at least one strictly negative utility $\exists i (u_i < 0)$, and

$$Q_{\text{IGM}}(\boldsymbol{h}, s, \boldsymbol{a}^*) = \underbrace{w(s, \boldsymbol{a}^*)}_{>0} \underbrace{f(u_1, \ldots, u_N)}_{<0} + b(s)$$
$$< b(s). \tag{51}$$

Therefore, $\boldsymbol{a}^* = \text{argmax}_{\boldsymbol{a}} Q_{\text{IGM}}(\boldsymbol{h}, s, \boldsymbol{a})$, and the actions that maximize the individual utilities also maximize the joint state-only value.

**Step 2.** Note that $\boldsymbol{a}^* = \text{argmax}_{\boldsymbol{a}} Q_{\text{IGM}}(\boldsymbol{h}, s, \boldsymbol{a})$ is valid for any state, at the very least because $\boldsymbol{a}^*$ are defined via the stateless individual utilities.

If $\boldsymbol{a}^*$ maximizes the joint history-state values for any given state, then it also maximizes the joint history-state values when marginalized over any distribution of state $p \in \Delta \mathcal{S}$, and $\boldsymbol{a}^* = \text{argmax}_{\boldsymbol{a}} \mathbb{E}_{s \sim p} [Q_{\text{IGM}}(\boldsymbol{h}, s, \boldsymbol{a})]$. This must be true also for the specific distribution $p(s) \doteq \Pr(s \mid \boldsymbol{h})$, and $\boldsymbol{a}^* = \text{argmax}_{\boldsymbol{a}} \mathbb{E}_{s|\boldsymbol{h}} [Q_{\text{IGM}}(\boldsymbol{h}, s, \boldsymbol{a})]$.

Therefore, the same actions $\boldsymbol{a}^*$ that maximize the individual utilities, also maximize the marginalized joint history-state values, satisfying the definition of state-based IGM in Definition 3.

$\square$

In contrast to history-state QFIX in Appendix E.1, we are not able to prove that state-only QFIX is able to represent the complete function class of IGM values.

## F EVALUATION DETAILS AND ADDITIONAL RESULTS

### F.1 SMACv2

**Implementation details** We note that `Pymarl2` provides *state-based* implementations of QMIX and QPLEX. For QPLEX in particular, this means that state-only weights $w_i(s)$ and $\lambda_i(s, \boldsymbol{a})$ are employed. As discussed by Marchesini et al. (2024), the state-only implementation of QPLEX loses some of the theoretical properties related to full IGM-completeness (and the same holds for Q+FIX, see Appendix E). However, to maintain a fair comparison, our implementation of Q+FIX employs analogous state-based implementation with state-only weights $w(s, \boldsymbol{a})$ for Q+FIX-{sum,mono}, and $w_i(s, \boldsymbol{a})$ for Q+FIX-lin. QPLEX and Q+FIX implementations both employ *advantage detaching* as previously described. For these SMACV2 experiments, we did not find it necessary to employ *intervention annealing*.

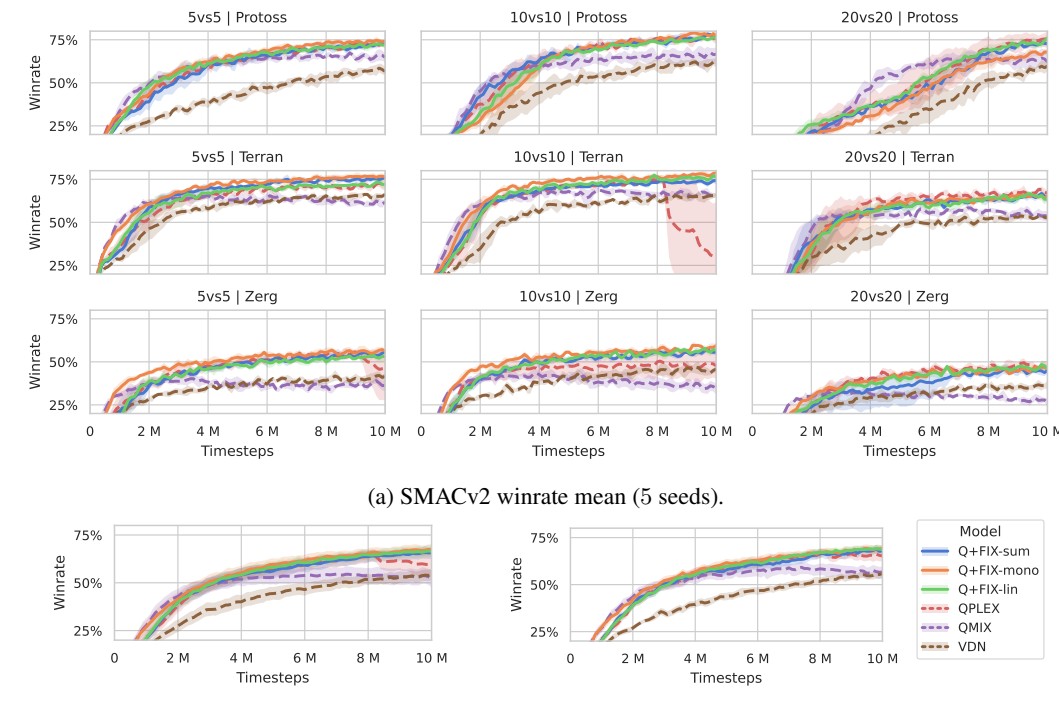

(a) SMACv2 winrate mean (5 seeds).

(b) SMACv2 winrate mean aggregate (45 seeds).     (c) SMACv2 winrate IQM aggregate (45 seeds).

Figure 5: SMACv2 winrate results, bootstrapped $95\%$ CI.

**Metrics**   SMACv2 logs various metrics pertaining to team performance, including the mean return and the mean winrate obtained as the ratio of episodes where the agents succeed in defeating the enemies. Although the winrate is a common metric used in prior work (e.g., Wang et al. (2020) use the winrate in their SMACv1 evaluation), we have found that winrates induce a different ordering over performances, i.e., it is possible to obtain a higher winrate while achieving a lower return, and vice versa. This indicates that the rewards of SMACv2 do not perfectly encode the task of defeating the enemies—a matter of reward design that is beyond the scope of this work. Since returns are the metric that the methods are directly trained to maximize, we prioritize returns as our primary evaluation metric in the main document, but also provide winrate results in this appendix.

**Winrate results**   In this section, we show additional results based on the winrate metric. As with the return-based results, we show the learning performance for each model and scenario in Fig. 5a, and the aggregate winrate across scenarios in Fig. 5b.

**Winrates vs returns**   As mentioned in the main document, the winrate and return metrics induce correlated but notably different orderings over the evaluated methods. Comparing Figs. 2 and 5, this is notable by the following (non-exhaustive) observations:

- In T5,
    - Return indicates Q+FIX-sum $\succ$ Q+FIX-mono.
    - Winrate indicates Q+FIX-sum $\prec$ Q+FIX-mono.
- In Z5,
    - Return indicates Q+FIX-sum $\succ$ Q+FIX-mono $\approx$ Q+FIX-lin.
    - Winrate indicates Q+FIX-sum $\approx$ Q+FIX-mono $\approx$ Q+FIX-lin.
- In Z10,
    - Return indicates VDN $\approx$ Q+FIX.
    - Winrate indicates VDN $\prec$ Q+FIX.

- In `P20`,
  - Return indicates VDN $\approx$ Q+FIX-mono.
  - Winrate indicates VDN $\prec$ Q+FIX-mono.
- In `T10`, the return of QPLEX drops significantly around the $9M$ timestep mark, whereas its winrate is able to recover temporarily, indicating that high winrates are achievable even with low returns.

Comparing the final performances in Figs. 2b and 5b,

- Return indicates VDN $\prec$ QMIX $\prec$ QPLEX.
- Winrate indicates QPLEX $\prec$ VDN $\approx$ QMIX.

**Winrate results discussion** Despite the notable differences between returns and winrates as evaluation metrics, the winrate-based evaluation arrives to largely the same conclusions as the return-based one in the main document, with respect to the performance evaluation of Q+FIX compared to other baselines.

As in the return-based results, VDN fails to be a competitive baseline on its own for most scenarios, likely due to the well-known limited representation. Fixing VDN via Q+FIX-sum, we are able to overcome this limitation (as noted by the performance gap between VDN and Q+FIX-sum), expanding its representation space and reaching SOTA performance.

As in the return-based results, QMIX sometimes exhibits fast initial learning speeds, albeit often to a sub-competitive final performance (`P5`, `T5`, `T10`, `Z10`, `T20`, `Z20`), again a likely consequence of its limited representation. Fixing QMIX via Q+FIX-mono, we are often able to exploit the initial learning speeds and complement them with improved performance at convergence reaching SOTA performance.

Compared to return-based results, QPLEX appears less competitive, and performs very well in fewer scenarios (`P20`, `T20`, `Z20`), and underperforms in more (`T5`, `Z10`), and exhibits the same troubling convergence instabilities as well (`Z5`, `T10`). Q+FIX-lin, as the simplified variant inspired by QPLEX, manages to avoid such convergence instabilities, plausibly as a consequence of the simpler structure.

As in the return-based results, Q+FIX-sum, Q+FIX-mono, and Q+FIX-lin achieve similar learning performances in most cases, with only minor differences across scenarios. Compared to the return-based results, it is Q+FIX-mono that may be slightly outperforming other variants in some scenarios (`T5`, `Z5`).

The aggregate results in Figs. 5b and 5c largely confirm the trends discussed above. Even when employing the IQM measure, which ignores the unstable QPLEX outlier rus, Q+FIX comes out as achieving higher performance. Despite the concerning difference between the return and winrate metrics, both demonstrate that Q+FIX succeeds in enhancing the native performances of VDN and QMIX fixees, and lifts them to a similar level as QPLEX while maintaining more stable convergence.

**Model size evaluation** One of the major appeals of Q+FIX over prior models is in its simplicity, and its ability to enhance prior models to achieve IGM-complete value function decomposition with small models. Because Q+FIX operates by augmenting existing fixee models with additional models $w(\boldsymbol{h}, \boldsymbol{a})$ and $b(\boldsymbol{h})$, there may be other concerns regarding whether the superior performance of Q+FIX comes simply as a consequence of the larger parameterization compared to the corresponding fixee. Table 2 contains a complete list of mixer sizes. Note that the mixer of Q+FIX-sum is always larger than that of VDN, and the mixer of Q+FIX-mono is always larger than that of QMIX. Therefore, there is a potential concern that the performance of Q+FIX (compared to its corresponding fixee) is driven by the additional parameterization rather than other factors like its proven theoretical properties.

In this section, we present an evaluation that disproves this concern by comparing the performance of a *bigger* fixee with a corresponding Q+FIX variant that employs a *smaller* fixee. We note that this evaluation is only possible for the case of QMIX and Q+FIX-mono: (i) VDN has no mixing network; therefore it is not possible to perform this evaluation for VDN and Q+FIX-sum. (ii) QPLEX is never

Table 2: SMACv2 mixer sizes in number of parameters. Smallest (non-zero) models highlighted.

| | Protoss | | | Terran, Zerg | | |
| | 5vs5 | 10vs10 | 20vs20 | 5vs5 | 10vs10 | 20vs20 |
|---|---|---|---|---|---|---|
| VDN | 0 k | 0 k | 0 k | 0 k | 0 k | 0 k |
| QMIX | 38 k | 83 k | 201 k | 36 k | 79 k | 194 k |
| QPLEX | 135 k | 326 k | 882 k | 126 k | 308 k | 846 k |
| Q+FIX-sum | 20 k | 50 k | 138 k | 19 k | 48 k | 133 k |
| Q+FIX-mono | 54 k | 180 k | 743 k | 50 k | 169 k | 708 k |
| Q+FIX-lin | 21 k | 51 k | 140 k | 19 k | 48 k | 135 k |
| QMIX-big | 166 k | 341 k | 767 k | 161 k | 331 k | 747 k |
| Q+FIX-mono-small | 29 k | 83 k | 290 k | 27 k | 78 k | 277 k |

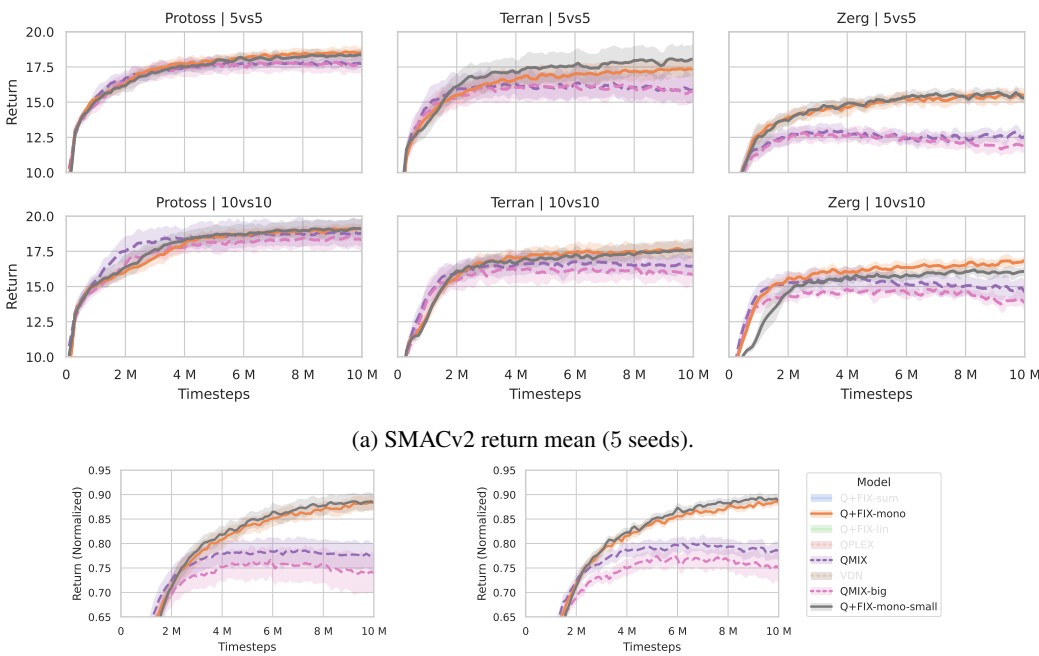

(a) SMACv2 return mean (5 seeds).

(b) SMACv2 return mean aggregate (30 seeds).  (c) SMACv2 return IQM aggregate (30 seeds).

Figure 6: SMACv2 model size results, bootstrapped $95\%$. Aggregation computed as in Fig. 2.

used as a fixee; therefore it is not possible to perform this evaluation for QPLEX (also, the Q+FIX models are all significantly smaller than QPLEX to begin with). Therefore, we implement a *bigger* variant of QMIX (QMIX-big) and a *smaller* variant of Q+FIX-mono (Q+FIX-mono-small). See in Table 2 that the size of Q+FIX-mono-small is now both smaller than that of QMIX-big, and more comparable to those of Q+FIX-sum and Q+FIX-lin.

**??** shows the results of this evaluation; to focus on the matter at hand, we only show the relevant performance of QMIX and Q+FIX-mono methods. As can be seen, the performance of Q+FIX-mono-small is analogous to that of +FIX-mono, and the performance of QMIX-big is analogous to that of QMIX. These results strongly confirm that the superior performance of Q+FIX-mono is not caused by the larger parameterization, but by our proposed fixing structure.

**Probability of improvement**    Agarwal et al. (2021) also suggest the use of *probability of improvement* (POI) as a criterion for evaluation that is resilient to data outliers. This metric measures the likelihood that a random run based on one method outperforms a random run based on another method, while ignoring the size of the performance gap. If method $X$ has been evaluated empirically $N$ times with performances $\hat{X} = \{\hat{x}_i\}_{i=1}^N$, and method $Y$ has been evaluated empirically $M$

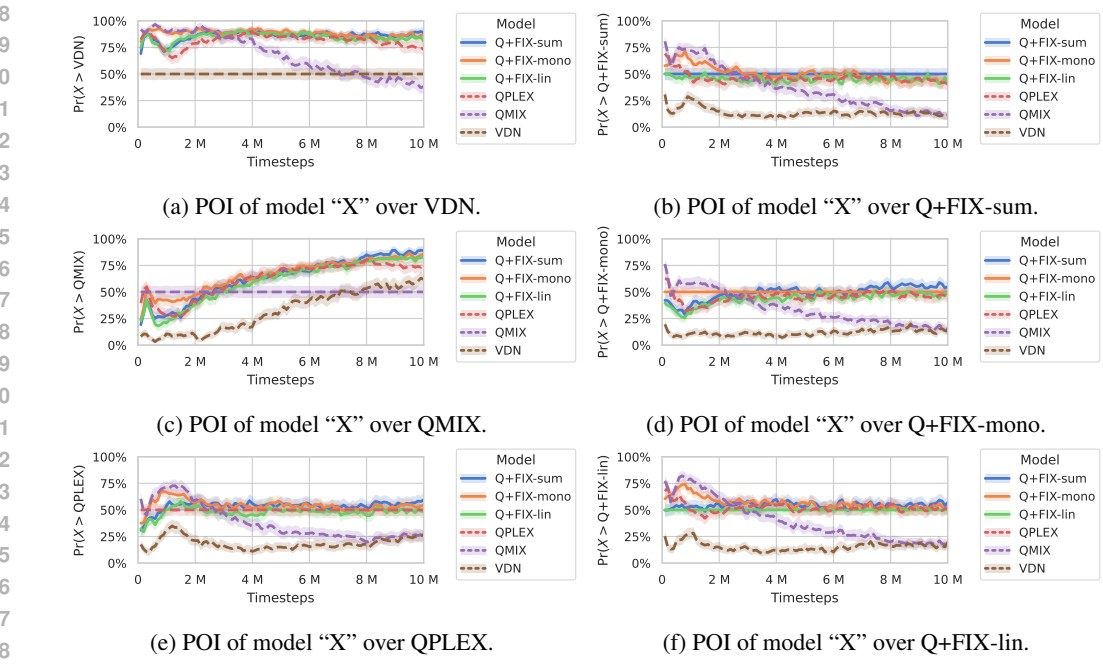

(a) POI of model "X" over VDN.

(b) POI of model "X" over Q+FIX-sum.

(c) POI of model "X" over QMIX.

(d) POI of model "X" over Q+FIX-mono.

(e) POI of model "X" over QPLEX.

(f) POI of model "X" over Q+FIX-lin.

Figure 7: Aggregate probability of improvement (POI), bootstrapped 95% CI.

times with performances $\hat{Y} = \{\hat{y}_i\}_{i=1}^M$, we estimate the POI as

$$\Pr(X > Y) \approx \frac{1}{N \cdot M} \sum_{\hat{x} \in \hat{X}, \hat{y} \in \hat{Y}} \mathbb{I}\left[\hat{x} > \hat{y}\right] . \tag{52}$$

In their work, Agarwal et al. (2021) demonstrate this criterion assuming that each run is summarized by a single scalar (e.g., final performance); since we are both concerned with learning speed and are uncertain how to fairly pick a single scalar performance for each run, we instead perform this calculation over the entire learning phase.

Fig. 7 contains our aggregate POI results for SMACv2. Agarwal et al. (2021) note that a POI that is above $50\%$ with its entire CI indicates a statistically significant result; out of all methods, Q+FIX-sum is the only one to achieve this against all other methods.

### F.2 OVERCOOKED

**Observability**   Overcooked is a fully observable environment, with each agent receiving observations whose information content is equivalent to the state. Therefore, the challenge of these tasks is primarily one of coordination and subtask assignment over information gathering. The state is provided as a tensor with shape $H \times W \times C$, with $C = 26$ (mostly but not exclusively binary) channels encoding agent positions and orientations, and positions of tables, pots, plates, various ingredients, etc.

**Coordination**   Notably, the tasks in overcooked do not strictly require tight coordination between agents. Though some tasks may need both agents to contribute in different ways to the same plate being completed, that cooperation is not under strict coordination requirements. Though the agents may achieve higher efficiency and performance if they coordinate optimally, the tasks can be completed even if the agents act relatively independently. We believe this can explain some of the results in our evaluation, especially in terms of the relatively good performance of methods like VDN that hardly enforce strong coordination.

**Implementation details**   For these Overcooked experiments, we found it useful for Q+FIX to employ both *advantage detaching* and *intervention annealing* with $\lambda$ descending linearly from 1 to 0 over the first $500k$ timesteps (10% of training).

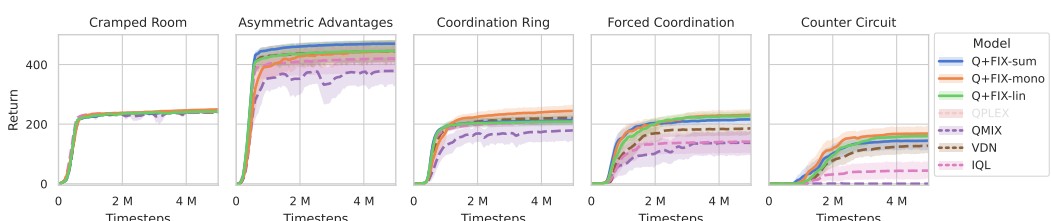

Figure 8: Overcooked return mean, bootstrapped 95% CI (20 seeds).

**Additional results** Fig. 8 shows the results for all five evaluated layouts: `Cramped-Room`, `Asymmetric-Advantages`, `Coordination-Ring`, `Forced-Coordination`, and `Counter-Circuit`. The tasks are categorically different and not directly comparable, but there is a progression in difficulty from the first to the last. `Coordination-Ring` is a simple task solved adequately by all methods. Performances start to differentiate more strongly in the other layout. Notably, QMIX has some trouble in these layouts, even compared to simpler methods like VDN. We believe this may be due to the coordination properties of these layouts (as described previously in this section), which may benefit simpler methods like VDN and independent learners. Nonetheless, in all scenarios, Q+FIX variants are the best performing, achieving statistically significant performance improvements compared to the baselines in four of the five layouts.

## G  ARCHITECTURES AND HYPERPARAMETERS

### G.1  SMACv2

Baseline methods are used and run as implemented in the `Pymarl2` repository[4], using the pre-optimized hyperparameters as provided by the corresponding configs. Q+FIX methods are implemented to match the baseline implementations completely, with the only difference being the mixer type and architecture, and are run using the same hyperparameters as the baselines. All implementations use the Adam optimizer Kingma & Ba (2017).

Due to their complex nature (including the use of hypernetworks and attention modules) we omit a full description of the mixing architectures for QMIX and QPLEX. We refer the reader to the corresponding publications and `Pymarl2` implementations[5,6].

**Agent Model** $\hat{Q}_i(h_i, a_i)$    All methods employ the same architecture to compute the individual utilities $\hat{Q}_i(h_i, a_i)$. As SMAXv2 is partially-observable and provides observations directly as feature vectors, this architecture employs the following layers:

- **Inputs:**
    - Observation vector $\mathbb{R}^d$ ($d$ variable per scenario).
    - Agent ID one-hot encoding $\{0, 1\}^N$.
- **Layers:**
    - `Linear(output_dim=64)` + `ReLU()`
    - `GRUCell(output_dim=64)`
    - `Linear(output_dim=#actions)`
- **Output:** Action values $\mathbb{R}^{|\mathcal{A}_i|}$, one per action.

---

[4]`https://github.com/benellis3/pymarl2`
[5]`https://github.com/benellis3/pymarl2/blob/master/src/modules/mixers/qmix.py`
[6]`https://github.com/benellis3/pymarl2/blob/master/src/modules/mixers/dmaq_general.py`

**Q+FIX Weight Model** $w(s, \boldsymbol{a})$

- **Input:**
  - State vector $\mathbb{R}^d$ ($d$ variable per scenario).
  - Agent actions one-hot encodings $\{0, 1\}^{\sum_i |\mathcal{A}_i|}$.
- **Layers:**
  - `Linear(output_dim=64)` + `ReLU()`
  - `Linear(output_dim=1)` (if Q+FIX-{sum,mono})
    `Linear(output_dim=N)` (if Q+FIX-lin)
  - `lambda w:  |w+1|-1+10e-8`
- **Outputs:** Weights $w(s, \boldsymbol{a}) \in \mathbb{R}_{>-1}$ (if Q+FIX-{sum,mono})
  Weights $w(s, \boldsymbol{a}) \in \mathbb{R}^N_{>-1}$ (if Q+FIX-lin).

**Q+FIX Bias Model** $b(s)$

- **Input:** State vector $\mathbb{R}^d$ ($d$ variable per scenario).
- **Layers:**
  - `Linear(output_dim=64)` + `ReLU()`
  - `Linear(output_dim=1)`
- **Output:** Bias $b(s) \in \mathbb{R}$.

### G.2 OVERCOOKED

Baseline methods are used and run as implemented in the `JaxMARL` repository[7], using the pre-optimized hyperparameters as provided by the corresponding configs. Q+FIX methods are implemented to match the baseline implementations completely, with the only difference being the mixer type and architecture, and are run using the same hyperparameters as the baselines. All implementations use the *rectified* Adam (RAdam) optimizer Liu et al. (2019).

**Agent Model** $\hat{Q}_i(h_i, a_i)$   All methods employ the same architecture to compute the individual utilities $\hat{Q}_i(h_i, a_i)$. As Overcooked is fully-observable and provides states as a grid (tensor) of categorical data, this architecture employs the following layers:

- **Input:** State grid $\mathbb{N}^{H \times W \times C}$ ($C = 26$ channels, mostly binary).
- **Layers:**
  - `Conv(output_dim=32, kernel_size=(5, 5))` + `ReLU()`
  - `Conv(output_dim=32, kernel_size=(3, 3))` + `ReLU()`
  - `Conv(output_dim=32, kernel_size=(3, 3))` + `ReLU()` + `Flatten()`
  - `Linear(output_dim=64)` + `ReLU()`
  - `Linear(output_dim=64)` + `ReLU()`
  - `Linear(output_dim=#actions)`
- **Output:** Action values $\mathbb{R}^{|\mathcal{A}_i|}$, one per action.

**Q+FIX Weight Models** $w(s, \boldsymbol{a})$

- **Input:**
  - State grid $\mathbb{N}^{H \times W \times C}$ ($C = 26$ channels, mostly binary).
  - Agent actions one-hot encodings $\{0, 1\}^{\sum_i |\mathcal{A}_i|}$.
- **Layers:**
  - `Conv(output_dim=64, kernel_size=(5, 5))` + `ReLU()`

---

[7]https://github.com/FLAIROx/JaxMARL

- Conv(output_dim=64, kernel_size=(3, 3)) + ReLU() + Flatten()
- Linear(output_dim=64) + ReLU()
- Linear(output_dim=1) (if Q+FIX-{sum,mono})
  Linear(output_dim=N) (if Q+FIX-lin)
- lambda w:  |w+1|-1+10e-8

- **Outputs:** Weights $w(s, \boldsymbol{a}) \in \mathbb{R}_{>-1}$ (if Q+FIX-{sum,mono})
  Weights $w(s, \boldsymbol{a}) \in \mathbb{R}_{>-1}^N$ (if Q+FIX-lin).

**Q+FIX Bias Model** $b(s)$

- **Input:** State grid $\mathbb{N}^{H \times W \times C}$ ($C = 26$ channels, mostly binary).
- **Layers:**
  - Linear(output_dim=64) + ReLU()
  - Linear(output_dim=1)
- **Output:** Bias $b(s) \in \mathbb{R}$.

## H  EXPERIMENTS COMPUTE RESOURCES

Experiments were distributed (unevenly) primarily across two workstations:

- **Type:** Standalone workstation,
  **CPU:** Intel(R) Core(TM) i7-7700K CPU @ 4.20GHz,
  **GPU(s):** 2x NVIDIA GeForce GTX 1080.
- **Type:** Standalone workstation,
  **CPU:** AMD Ryzen Threadripper 7960X 24-Cores,
  **GPU(s):** 1x NVIDIA GeForce RTX 4090.

The time of executing a single run can differ greatly depending on the workstation, the environment, the method, and model size. The following is only a very rough estimate of total sequential runtime:

**SMACv2** Pymarl2 implementations can be very slow due to the CPU-bound environment, and vary somewhere between $5\,\mathrm{h}$ and $20\,\mathrm{h}$ per run. For the main results, since we execute 6 methods for 5 runs in 9 scenarios, which amounts to $6 \cdot 5 \cdot 9 = 270$ independent runs and roughly $270 \cdot 12\,\mathrm{h} = 135\,\mathrm{d}$ of sequential runtime. For the results on model sizes, we execute an additional 2 methods for 5 runs in 6 scenarios, which amounts to $2 \cdot 5 \cdot 6 = 60$ independent runs and roughly $60 \cdot 12\,\mathrm{h} = 30\,\mathrm{d}$ of sequential runtime.

**Overcooked** JaxMARL implementations are much faster, and vary between $15\,\mathrm{min}$ and $60\,\mathrm{min}$. Since we execute 6 methods for 20 runs in 5 layouts, which amounts to $6 \cdot 20 \cdot 5 = 600$ independent runs and roughly $600 \cdot 40\,\mathrm{min} = 24\,000\,\mathrm{min} \approx 16\,\mathrm{d}$ of sequential runtime.

Naturally, the experiments were not executed purely sequentially; however, they still took multiple weeks to complete as a whole, on our available hardware.

