# OpenReview forum: "Fixing Incomplete Value Function Decomposition for Multi-Agent Reinforcement Learning"
_ICLR.cc/2026/Conference — Submitted to ICLR 2026_

### Official Review · Reviewer_W1FR · 2025-10-21

**Soundness:** 2
**Presentation:** 2
**Contribution:** 3
**Rating:** 4
**Confidence:** 5

**Summary:**

This paper investigates the problem of complementing value-decomposition methods which are not capable of achieving the complete class of IGM value-functions. To do so, it first proposes a simplified version of QPLEX IGM criteria, which are then in turn adapted into a fixing mechanism for other existing value-decomposition methods. Furthermore, a more practical version of the method is derived, which holds the same theoretical guarantees. Experimental results show how the fixed versions of various value-decomposition methods achieve better performances and stability than their original counterparts.

**Strengths:**

Given the extreme popularity of value-decomposition methods, and their known limitations, the proposed methodology that aims at fixing them in a clear and simple way is a good contribution to the field. The discussion of why QPLEX, although in principle capable of representing the complete IGM class, is defined redundantly or sometimes ill-posed is very clear and interesting, and it serves as a basis for the simpler formulation that is then used in proposing QFIX. I also liked the results on the state-based variant, which gives consideration to an aspect that is often overlooked in CTDE. Experimental results on the proposed problems are very good, and show how the fix can help in improving learning performances.

**Weaknesses:**

The main weakness of the paper is in its empirical evaluation. Although, as I listed in the strengths, the proposed results are good indeed, what it is missing here in my opinion is some rigour. Only the Q+FIX variant is evaluated, but no results for the original QFIX formulation is provided. This makes it difficult to assess how much better Q+FIX is in practice, and if the two are also equivalent in terms of practical performances. Also, the investigated problems are not known for being un-learnable with existing value-decomposition methods, but only to be difficult. For example, some simpler problem, but with an accent on the actual representation limitations of these, would have been much more interesting to prove the impact of your proposed fixing beyond a "simple" improvement in performances.

**Questions:**

- Why do you say the WQMIX is specifically developed for fully observable settings? The original paper uses Dec-POMDP as its working setting, and all the practical formulations and empirical results are all proposed on the partially observable setting.

- The beginning of Section 4.1 is not very explicative: while I can follow your reasoning here, having a more detailed explanation of both why three of the four constraints are satisfied by definition and why and how there is a misspecified case would help a lot in clearly understanding the problems you are highlighting with QPLEX.

- Perhaps explain what you mean with *measurable* IGM values would help in better understanding the properties of your proposed QFIX.

- Figure 1 is a bit difficult to read, as the text and details are fairly small. Given that the two figures are basically identical except for how $w(\mathbf{h},\mathbf{a})$ is computed, perhaps simply merge the two into a single figure which highlight such difference?

- Why not evaluating QFIX as well? It would have been interesting to assess the learning performance gap between this and its more learning-amenable variant Q+FIX, while this way we have basically no clue but your own words to believe that Q+FIX is more suitable for learning than the base QFIX formulation.

- It would have been interesting to show results on a setting where existing value-decomposition methods like VDN and QMIX are known to fail, like the popular climb matrix game. If their fixed counterparts were capable of achieving the optimal policy there, then this would have been a clear indication that the fixing process is indeed extending their representation capabilities. The current experiments are too large and complex to be able to say anything too precise on its effect per-se, other than the fixes are helping in learning faster and better generally.

- Your statement that your proposed QFIX requires smaller mixing models is not really true: while Q+FIX-lin is indeed smaller than QPLEX, getting rid of all the non-required components that you highlighted in your analysis, Q+FIX-mono is using more parameters than QMIX alone, and Q+FIX-sum is using some where standard VDN do not. I suggest you reformulate such a statement to be closer to what the situation actually is.

---

> ### Author Response · Authors · 2025-11-21
>
> We thank the reviewer for the constructive feedback.
>
> - **re: the observability assumptions employed by WQMIX**, while the original paper introduces Dec-POMDPs and claims to propose a method that is applicable to Dec-POMDPs, the actual notation, derivations, and proofs of WQMIX are all formulated exclusively under full observability, i.e., in reference to states, rather than joint/individual histories.  None of the claims in the paper are actually verified for the case of decentralized control, and we believe that they are unverifiable. As a simple example, consider equation (1), which defines the joint Bellman optimality operator and is the basis for the entire derivation. This equation defines the **centralized** Bellman operator, which has a fixed point that encodes the optimal behavior of a **centralized** team of agents. The theory of WQMIX claims to be able to extract the same optimal actions as those obtained by the fixed point of this centralized Bellman operator; if this were true even in the decentralized control case (for which no such Bellman operator is clearly identified by the literature), that would prove that all centralized behaviors can be executed in a decentralized manner, which is trivially incorrect. Somewhere in the derivation and proofs of WQMIX, full observability is explicitly assumed and plays a much bigger role than what is claimed in the paper.
>
> - **re: section 4.1**, we will expand on the explanation in the final version of the manuscript to clarify these points.
>     - Proposition 1, equation 4, lhs, is automatically true by the definition of $a^*$, which is a joint action that maximizes the joint $Q$ (and $A$) functions.
>     - Proposition 1, equation 5, lhs, is automatically true by the definition of $a$, which is a joint action that **does not** maximize the joint $Q$ (and $A$) functions.
>     - Proposition 1, equation 5, rhs, is automatically true because all individual advantages are non-positive by definition.
>     - The misspecification exists in that IGM forbids situations where the joint $A$ is non-zero when all the individual $A_i$ are zero, while Proposition 1 allows for such a situation.
>
> - **re: measurable functions**, we will add a brief explanation or reference to the concept of measurable functions as suggested. This is a property important for measure theory and function approximation, and a common assumption employed by universal approximation theorems; it is related (among other things) to a function having a number of discontinuities that can still be handled and approximated according to a given metric. Its relevance is due to the fact that universal approximation theorems are always formulated based on some kind of continuity of measurability assumptions, and no appeal to the UAT (as is the case in the QPLEX paper) is complete without making those same assumptions. In principle, this means that there may exist a whole subset of IGM but non-measurable functions that cannot be represented by QPLEX or QFIX, though this limitation is largely theoretical and unlikely to be relevant in practice, as those are simply functions that cannot be approximated in general by computational models.
>
> - **re: figure 1**, the way that the final value is combined is also different between QFIX and Q+FIX. We believe that overlapping the figures would make it harder to distinguish between the two methods. If readability is a concern (only really true for paper-prints), we may consider increasing the sizes of the figures in the final version of the manuscript to improve readability, once we have access to an additional page.
>
> - **re: evaluation of QFIX**, the mentioned claims are not about Q+FIX performing well (we have the empirical evaluation to support that), they are on QFIX performing poorly. QFIX alone performed too poorly on its own in practice to justify the use of scarce computational resources on a fuller evaluation suitable for publication, rather than extending the empirical evaluation to more environments; this is what motivated the whole derivation of Q+FIX in the first place. This indicates that the detachment of advantages plays an important and yet not entirely understood role in stabilizing learning for methods like QPLEX and Q+FIX alike.
>
> - **re: evaluation on simpler matrix games**, we will work on extending the evaluation to include such simpler games.
>
> - **re: claims of smaller models**, since QFIX is a wrapper around a fixee model, it is naturally true that QFIX is never smaller than the fixee it employs. Most of our claims explicitly mention that our models are **smaller than QPLEX**, the main other IGM-complete competitor. Related to this topic, the appendix also contains a further ablation where we confirm that a Q+FIX-mono that employs a small QMIX fixee performs better than a bigger QMIX model alone. We are happy to further clarify all other statements likewise to avoid any possible confusion in the final manuscript.

---

> > ### Comment · Reviewer_W1FR · 2025-11-23
> > **Reply to Authors**
> >
> > I would like to thanks the authors for their useful comments. Here are some additional points below:
> >
> > - WQMIX: I see your point, and it is fair. However, the algorithm as in the main paper is formulated to work under partially observable settings as well. I found your claim a bit imprecise: please clarify that you are referring to the *theoretical guarantees* of WQMIX rather than to the algorithm applicability itself.
> >
> > - QFIX evaluation: while I see the point of QFIX being weak in practice, this is something that does not come across in the shown empirical results, and that I think it would be fair to present and discuss, so that the reader can 1. Give a better reason for Q+FIX to be proposed and used and 2. Clearly see this point, for clarity and honesty.
> >
> > - Matrix game: what I meant here is not simply to add some results on a specific problem, but rather to show QFIX/Q+FIX benefits on problems where the incompleteness of the IGM factorization really plays a role. In problems like SMAC it is difficult to assess if poor performances of a given algorithm are due to incomplete factorizations or other compounding factors, so to me your results show that Q+FIX is a better learning algorithm but not specifically better at complementing IGM factorizations. Using simpler problems where we know that the bad performances of some algorithms are really due to problems with representing a factorizable function, then QFIX/Q+FIX eventual success would really tell me what you are claiming that it complements the IGM representation to address other methods' limitations.
> >
> > - Smaller models: yes, I know that being an additional wrapper around existing methods it cannot have less parameters than its unfixed counterpart, but this is not explained clearly enough in the paper in my opinion.

---

### Official Review · Reviewer_SVj3 · 2025-10-31

**Soundness:** 3
**Presentation:** 3
**Contribution:** 3
**Rating:** 6
**Confidence:** 3

**Summary:**

This paper addresses the representational limitations of common value-decomposition methods in cooperative MARL (e.g., VDN, QMIX) under the individual-global-max (IGM) principle. The authors argue that while methods like QPLEX are provably IGM-complete, they are overly complex. They propose a principled and simpler formulation of IGM-complete value decomposition and introduce QFIX, a family of methods that “fix” incomplete decompositions by adding a thin correction layer. An additive variant (Q+FIX) is shown to significantly improve stability and performance. Experiments on benchmarks, such as SMACv2, demonstrate that Q+FIX improves over VDN/QMIX and matches or exceeds QPLEX while being simpler and smaller.

**Strengths:**

1. The paper provides a clean, minimal characterization of the IGM-complete function class and highlights a core mechanism (weighted transformation of advantages) without complex transformation stacks.
2. QFIX uses a “fixing” layer over existing factorizations, requiring small architectural changes.
3. Benchmarks across SMACv2 (9 scenarios, multiple races/sizes) and Overcooked; includes stability and model-size comparisons. The proposed method exhibits better convergence stability than QPLEX and uses fewer parameters.

**Weaknesses:**

1. Comparisons only within value-decomposition class (no MAPPO, MADDPG variants).
2. Some practical tricks (advantage detach, annealing) feel heuristic, and they seem to lack a limited theoretical justification.
3. Limited analysis of when fixing hurts performance.

**Questions:**

1. How sensitive is QFIX/Q+FIX to hyperparameters relative to QMIX/QPLEX?
2. Could fixing networks destabilize learning in extremely large action spaces?
3. The baseline algorithms selected are necessary but old. How is the performance when compared against newer algorithms, such as ResQ, or other types of methods like MAPPO?

---

> ### Author Response · Authors · 2025-11-21
>
> We thank the reviewer for the constructive feedback.
>
> - **re: sensitivity to hyperparameters**, we were unable to perform an extensive sensitivity study on hyperparameterization due to resource limitations, and decided to prioritize our limited computational resources into obtaining results in a more diverse set of environments. Whenever appropriate, our QFIX/Q+FIX variants are based on the same parameterizations as the corresponding fixees. This appears to be sufficient, though it does open the possibility that further hyperparameter tuning could further improve the performance of QFIX/Q+FIX by making different choices.
>
> - **re: could fixing networks destabilize learning in extremely large action spaces, and when does fixing hurt performance**, this is an interesting but hard question to answer definitively without further empirical evidence focused on large action spaces. However, we note that the effects of the fixing networks $w(h, a)$ and $b(h)$ have a fairly simple effect on the outputs; $b(h)$ does not depend on actions at all, and does not affect action selection, so we do not expect it to have a destabilizing effect based on the complexity of the action space. The $w(h, a)$ network does depend on actions, but it is a mere multiplicative factor on the total advantage of the fixee; as such, the learned models can always easily stabilize around the "do-nothing" value of $w(h, a) = 1$, even in the worst case. This is possible for complexity that may render fixing too complex, including large action spaces.
>
> - **re: comparison exclusive to value-decomposition class of methods**, we agree that it would be interesting to compare with such methods, and that more evaluation is always welcome. However, the focus of this work is on value-decomposition methods, their representational limitations, and their parameterization. An evaluation of whether value-function-decomposition as a whole is a better or worse approach than policy-gradient methods in CTDE settings is an orthogonal research question that we do not aim to address in this work. Ultimately, we are limited by computational resources and research scope, and are unable to perform an exhaustive evaluation of all possible baselines; we are limited to a selection of baselines that are most relevant to the core contributions of this work.
>
> - **re: the heuristic nature of implementation details**, we agree that implementation details like the advantage detachment and fixing annealing are heuristic in nature, and deserve a closer look in future work. We highlight that advantage detachment was already employed by QPLEX, and our evaluation has further demonstrated its necessity. We are also the first to formulate a concrete hypothesis on why it may be important (Appendix D), though more analysis is certainly warranted.

---

### Official Review · Reviewer_JrsL · 2025-10-31

**Soundness:** 2
**Presentation:** 3
**Contribution:** 2
**Rating:** 2
**Confidence:** 4

**Summary:**

This paper proposes QFIX, a new method to address representational constraints imposed by some methods that adhere to the IGM condition, such as VDN and QMIX. This method uses a fixing layer to ensure that it can represent the full classes of functions that satisfy the IGM condition. This layer can be applied ot other existing methods and extend their representational capabilities.

**Strengths:**

This paper is well written and easy to read. The experiments provided are quite extensive and the method has strong theoretical groundings. The authors also aim to address specific limtiations of other previous methods, which is good.

However, there are some points that are a bit less clear. Please find below.

**Weaknesses:**

Overall, I am unsure about the strength of the motivations for the proposed method; the proposed method is based on a structure that contains all functions that satisfy IGM, but other methods such as QTRAN can theoretically factorise any function which means that all functions that satisfy IGM are theoretically also included in that set of functions. QPLEX also has a strong representational complexity and the argument that it is "unnecessarily complex" does not sound convincing enough as a motivation without some further empirical support, for instance. In addition, the authors build the argument around other methods not being able to represent all functions that satisfy IGM which is valid. However, methods such as QTRAN have theoretically proved to be able to factorise any family of functions, which implies that they are able to factorise all functions that satisfy IGM.

The authors claim that the necessity some the architectural choices of QPLEX such as some transformations is questionable; while that seems valid from a theoretical perspective in terms of being IGM-complete, it is not clear whether removing these transformations will also decrease their performance, meaning that they might indeed be necessary in practice.

The performance of the proposed method is very close to the baselines in almost every SMAC scenario, which makes me question the necessity for such approach in terms of practical performance.

I could not find some of the proofs in the paper, for example for propositions 1 and 2; if they are taken from other works these should be mentioned accordingly.

Minor:
- the meaning of some terms in equations could be better defined; for example, $Q_-$ in eq 1 and "hao"

[1] QTRAN https://arxiv.org/abs/1905.05408

[2] QPLEX https://arxiv.org/abs/2008.01062

[3] DuelMIX: https://arxiv.org/pdf/2408.15381

**Questions:**

1. why is it more advantageous to be IGM-complete (containing only all the functions that satisfy IGM, as per Def 2) instead of being able to factorise any function, such as in the case of QTRAN?
2. how do the authors ensure that (i) is satisfied? i.e., avoid assumptions like full observability or centralised control (lines 49-51)
3. since this work builds upon some of the insights from DuelMIX [3], it would be useful to see how the proposed method compares to DuelMIX, both theoretically and in practice
4. in figure 1 and in the equations (for example eq 10) the authors use the notation $V(h)$ similarly to QPLEX notation of $V(\tau)$, i.e., the function is based on the history; however, in [3] it is argued that in QPLEX this theory is inconsistent with the practice where the authors use the state instead, which can cause inconsistencies and degradation in learning. Is that the case in this work as well?
5. the authors propose 3 different variations of QFIX (sum, mono and lin) which in essence the differences seem to be in the mixer; would it be possible to generalise everything in a single method that could fix all the baseline methods mentioned at once? also how easy would it be to apply this method to other value function factorisation methods since apparently a distinct one is defined for each basline?
6. it is unclear to me the intuition for proposing QFIX and then Q+FIX right after if only Q+FIX is used in the experiments as the proposed method; why not only introducting Q+FIX and building from the equations of QFIX but in the same section?

---

> ### Author Response · Authors · 2025-11-21
>
> We thank the reviewer for the feedback, though we believe the reviewer is mistaken on some core aspects of value-function decomposition (including QTRAN and DuelMIX), misunderstanding some aspects of our proposed formulation, as well as how our evaluation is confirming our claims about prior work.
>
> - **re: representational capabilities of QTRAN**, the claim that QTRAN is able to factorize arbitrary functions , not just the IGM space, is incorrect; Firstly, QTRAN is limited to modeling behaviors compatible with IGM, as also discussed in the corresponding publication. Secondly, the representation capability of QTRAN is more narrow than the IGM space; what QTRAN does is build an additive VDN-style model (with limited representation) whose only guarantee is that it is associated with the same maximal action as a given IGM-compliant value. That is not the same as being able to perform IGM-complete factorization; QTRAN can at best build a model that has the same argmax as Q, with no guarantees about the values; e.g., QTRAN cannot model all of $Q$=(1, 2; 2, 4), but it can at best define a model whose maximal action is in the bottom-right one. In contrast, proper factorization entails being able to model all values, which has other benefits like credit assignment.
>
> - **re: the potential importance of the QPLEX transformations**, our empirical evaluation addresses this point directly, as the performance of QFIX confirms that the QPLEX transformations are not practically necessary, and may be the cause of learning instabilities.
>
> - **re: Props 1 and 2**, the reference for Prop 1 is in the text just prior; we will clarify this in the updated version. Prop 2 is presented as a direct variant of Prop 1, that addresses the issue described just before it. Its proof is the same as that of Prop 1, (from the referenced work); we are merely fixing the inaccuracy in the statement of Prop 1 itself. We are happy to add explicit proofs in order to better differentiate Props 1 and 2.
>
> - **re: avoiding assumptions of full observability/centralized control**, these assumptions are avoided by adhering to the IGM condition rigorously, and by not conflating joint histories with states, (e.g., as opposed to WQMIX).
>
> - **re: connection to DuelMIX**, the connection between QFIX and DuelMIX is only relevant in the context of models that also employ state information, which is not a primary aspect of QFIX or its derivation. We separate these two topics specifically by deriving QFIX in the “purer” state-less context, and then providing a separate complementary analysis of the state-ful case in the appendix, comparable to the analysis of prior work from the DuelMIX paper. The state-ful mixing case is not the main theoretical focus of this work, but being the case that it is a common practical implementation detail of many methods (even when the theory does not reflect it), we felt that it was important to address it via analysis and evaluation.
>
> - **re: role of state in QFIX**, in order to maintain a fair evaluation with respect to prior work, we derive theoretical results for both the state-less and state-ful cases. In the practical evaluation, because other baselines are often implemented exclusively in the state-ful case, we also implement QFIX similarly, to avoid compromising the results by introducing discrepancies in the type and amount of information that is available to the different methods.
>
> - **re: generalizing QFIX-{sum,mono,lin}**, QFIX in equation (11) is already the generalization you are asking for; QFIX-{sum,mono,lin} are specializations of QFIX, shown explicitly in equations (13, 14, 15) to aid clarity on how these look in practice. The assumptions of our method state that we can apply QFIX to fix any IGM-incomplete method, with QFIX-{sum,mono,lin} being just three natural examples. There is no intrinsic limit to what method may be employed as a fixee, though it makes little sense to employ a method that is already IGM-complete.
>
> - **re: distinction between QFIX and Q+FIX**, the reason to introduce QFIX is that it encodes the IGM property more clearly and directly. Equations (8, 11) are a direct mathematical translation of the statement that "any deviation from individual maximality can be translated to an arbitrary deviation from joint maximality", which describes the IGM condition and IGM space. Q+FIX is a practical reparameterization of QFIX, that enables the use of implementation details that appear to be important for stable learning. We do not show results for QFIX itself because it performs poorly, demonstrating the importance of those implementation details (even though QFIX and Q+FIX are equivalent in terms of representation).
>
> - **re: minor points**, we will clarify this in the paper.
>     - $Q^-$ in value-based methods often denotes a slowly-updated target value used to compute stable TD targets.
>     - $hao$ denotes the updated history resulting from a previous history $h$, action $a$, and new observation $o$.

---

### Official Review · Reviewer_sZyZ · 2025-11-01

**Soundness:** 4
**Presentation:** 3
**Contribution:** 4
**Rating:** 8
**Confidence:** 4

**Summary:**

The paper proposes a characterization of the IGM-consistent function class using advantage-level equivalences and uses it to derive QFIX, which wraps any IGM but a non-IGM-complete decomposer with a family of value function decomposition models based on it.

**Strengths:**

- I find the theoretical contribution sound and ample. The proposal of QIGM is compact and complete with proof constructive. I find most of the theoretical derivation correct.

- The measurable-UAT framing is accurate and improves the rigorisness of IGM-complete in literature.

- The fixing intervention that empirically stabilizes and improves performance verified in the experiments.

- The experiment design is carefully designed with comprehensive benchmarks and representative baselines for theoretical improvement demonstation.

**Weaknesses:**

- The measurable-UAT claim is correct but depends on appendix-level assumptions. A short brief mention would complete the main text during network family explanation and convergence.

- Missing some of the SOTA baselines like MAPPO or more recent works, but given this experiment design is more on verifying the theoretical contribution I think this is acceptable to only compare with QMIX and QPLEX.

- This paper could also benefit from simplifying the key assumption and findings and leave most of the theoretical proof to appendix.

- Some very minor presentation issue like missing ref on line 1178

**Questions:**

In Overcooked, Q+FIX benefits from annealed intervention in addition to detach. I wonder why this simple design can help the stabilizers to all baselines in a controlled way.

Parameter-count comparisons focus on mixers; indicating agent parameters are the same? Also time-to-target metric would better disentangle capacity vs. architecture effects. Some size controls are present but could be expanded.

---

> ### Author Response · Authors · 2025-11-21
>
> We thank the reviewer for the constructive feedback.
>
> - **re: measurable-UAT assumptions**, we agree that the assumptions required for the measurable UAT to hold are important for a more formal treatment of the theoretical results, and we will add a brief mention of them in the main text when we first introduce the concept, as suggested.
>
> - **re: the lack of baselines from policy-gradient methods**, we agree that it would be interesting to compare with such methods, and that more evaluation is always welcome. However, as the focus of this work is on value-decomposition methods, their representational limitations, and their parameterization, we believe that comparing with other value-decomposition methods (both IGM-incomplete and IGM-complete) is sufficient to demonstrate the effectiveness of our proposed method. The topic of whether value-function-decomposition as a whole is better or worse than policy-gradient methods in CTDE settings is an orthogonal research question that we do not aim to address in this work.
>
> - **re: the annealed intervention**, the annealed intervention can only be implemented in baselines that apply an additive "residual" modification to the fixee's joint values, which excludes both VDN and QMIX. In principle this could potentially be applied to QPLEX in the SMACv2 benchmarks (as it is possible to rewrite the QPLEX formulation in an additive form as well), but we found no need to apply the annealed intervention in the SMACv2 experiments to achieve good performances.
>
> - **re: parameter-count comparison**, we confirm that the agent model parameterization is the same for all methods, hence our focus on the parameterization of the mixers alone.
>
> - **re: the time-to-target metric**, while time-to-target is indeed a valuable metric generally speaking, we believe that it is redundant and less informative in our evaluation.
>     - It is redundant because the time-to-target can be read from our learning curves in figures (2, 3, and others in the appendix) by taking a horizontal cross-section of each learning curve and finding the earliest point where the learning curves cross the horizontal section.
>     - It is less informative because it fails to capture the instability of methods whose performance grows to a certain performance before collapsing to a lower value (as demonstrated in some cases by QPLEX). Hence we cannot focus exclusively on the earliest time when a given method achieves a target performance.
>
> We also appreciate the reviewer pointing out the issue with the missing reference in line 1178, which we will fix in the final version.

---

### Meta-Review · Area_Chair_pk5z · 2026-01-03

**Summary:**

This work finds that multiple existing works (e.g., VDN and QMIX) cannot represent the full class of IGM values, and QPLEX is unnecessarily complex. It proposes QFIX, which wraps any IGM but a non-IGM-complete decomposer with a family of value function decomposition models based on it. QFIX uses a simple “fixing” network to extend the representation capabilities of prior methods. The authors show that the proposed approach, QFIX, is promising on SMACv2 and Overcooked.

The strengths of QFIX are listed as follows.
1. The paper is well written.
2. QFIX uses a “fixing” layer over existing factorization methods, requiring small architectural changes.

The weaknesses of QFIX are listed as follows.

1. The claim that "the one exception that has no such limitation (QPLEX) is unnecessarily complex" is overclaimed. For example, WQMIX, QTRAN can model MARL function without representation limitations.

2. Incomplete related work on MARL value function decomposition. The related work section discusses work published before 2021, which is 4-5 years ago. Multiple value function decomposition methods are not compared through experiments.

3. As it is written by one reviewer, the authors do not demonstrate the QFIX/Q+FIX benefits on problems where the incompleteness of the IGM factorization really plays a role.

4. As it is written by one reviewer, the benefits of Q+FIX over QFIX do not shown across all the experiments.

5. The result figures should be improved for better readability.

**Reviewer Concerns:**

1. The authors do not demonstrate the QFIX/Q+FIX benefits on problems where the incompleteness of the IGM factorization really plays a role.

2. The benefits of Q+FIX over QFIX do not shown across all the experiments.

3. Some practical tricks (advantage detach, annealing) feel heuristic, and they seem to lack a limited theoretical justification.

**Reviewer Scores:**

The authors do not respond to the last comments of reviewer W1FR. The concerns are not addressed.  The authors can respond to the review and comment in openreview till Dec 3. However, they did not reply to the reviewer's concerns posted in Nov 23.

---

### Decision · Program_Chairs · 2026-01-26

Reject